# Sox9 and Rbpj differentially regulate endothelial to mesenchymal transition and wound scarring in murine endovascular progenitors

Jilai Zhao [1,6], Jatin Patel[1,2,6], Simranpreet Kaur[1], Seen-Ling Sim[1], Ho Yi Wong[1], Cassandra Styke [1], Isabella Hogan[1], Sam Kahler [1], Hamish Hamilton[1], Racheal Wadlow[1], James Dight[1], Ghazaleh Hashemi [1], Laura Sormani[1], Edwige Roy [1], Mervin C. Yoder [3], Mathias Francois[4,5] & Kiarash Khosrotehrani [1✉]

Endothelial to mesenchymal transition (EndMT) is a leading cause of fibrosis and disease, however its mechanism has yet to be elucidated. The endothelium possesses a profound regenerative capacity to adapt and reorganize that is attributed to a population of vessel-resident endovascular progenitors (EVP) governing an endothelial hierarchy. Here, using fate analysis, we show that two transcription factors SOX9 and RBPJ specifically affect the murine EVP numbers and regulate lineage specification. Conditional knock-out of *Sox9* from the vasculature (*Sox9fl/fl/Cdh5-CreER RosaYFP*) depletes EVP while enhancing *Rbpj* expression and canonical Notch signalling. Additionally, skin wound analysis from *Sox9* conditional knock-out mice demonstrates a significant reduction in pathological EndMT resulting in reduced scar area. The converse is observed with *Rbpj* conditionally knocked-out from the murine vasculature (*Rbpjfl/fl/Cdh5-CreER RosaYFP*) or inhibition of Notch signaling in human endothelial colony forming cells, resulting in enhanced *Sox9* and EndMT related gene (*Snail, Slug, Twist1, Twist2, TGF-β*) expression. Similarly, increased endothelial hedgehog signaling (*Ptch1fl/fl/ Cdh5-CreER RosaYFP*), that upregulates the expression of *Sox9* in cells undergoing pathological EndMT, also results in excess fibrosis. Endothelial cells transitioning to a mesenchymal fate express increased *Sox9*, reduced *Rbpj* and enhanced EndMT. Importantly, using topical administration of siRNA against *Sox9* on skin wounds can substantially reduce scar area by blocking pathological EndMT. Overall, here we report distinct fates of EVPs according to the relative expression of *Rbpj* or Notch signalling and *Sox9*, highlighting their potential plasticity and opening exciting avenues for more effective therapies in fibrotic diseases.

[1] The University of Queensland Diamantina Institute, The University of Queensland, Woolloongabba, QLD, Australia. [2] Centre for Ageing Research Program, Queensland University of Technology, Woolloongabba, QLD, Australia. [3] Indiana Center for Regenerative Medicine and Engineering, Indianapolis, IN, USA. [4] The David Richmond Laboratory for Cardiovascular Development: Gene Regulation and Editing Program, The Centenary Institute, Camperdown, NSW, Australia. [5] The School of Life and Environmental Sciences, Faculty of Science, The University of Sydney, Camperdown, NSW, Australia. [6]These authors contributed equally: Jilai Zhao, Jatin Patel. ✉email: k.khosrotehrani@uq.edu.au

During adult life the endothelial layer that lines the lumen of the vascular system is heterogeneous in phenotype and retains the capacity to drive tissue regeneration and repair with responsive adaptations to both physiological and pathological conditions[1,2]. However, mechanisms that drive endothelial activity and plasticity, in particular those that contribute to fibrotic disease associated with adult wound healing through a process of endothelial to mesenchymal transition (EndMT) remain to be elucidated[3,4]. In prenatal development, EndMT is essential in endocardial cushion formation, cardiac fibroblasts and smooth muscle cell generation[5,6]. By contrast, in the adult vascular network, EndMT heavily influences the pathogenesis of several fibrotic diseases and is an essential process during wound healing[7]. In the cardiac setting, Zeisberg et al., (2007) demonstrated that EndMT played a significant role in the formation of excessive cardiac fibrosis, in situations of cardiac injury and subsequent repair[8]. Endothelial cells (ECs) adopted a fibroblast-like fate, becoming more proliferative, thrombogenic and expressed large amounts of extracellular matrix (ECM) proteins, resulting in cardiac fibrosis and driving disease progression. Similarly, during cutaneous wound healing, after a peak in neo vessel formation, ECs undergo EndMT and contribute to scarring[9]. These examples highlight the importance of endothelial plasticity and cell fate decision during health and disease. During wound healing, the endothelium contributes to the expansion of the vasculature by adopting endothelial phenotypes and later, transitioning towards a mesenchymal phenotype, it contributes to fibrosis using yet unclear mechanisms.

Recently, we and others demonstrated that a tissue resident endovascular progenitor (EVP) exists throughout the circulatory system, which governs an endothelial hierarchy[10–12] that leads to de novo blood vessel formation. During homoeostasis and disease, quiescent EVPs give rise to rapidly proliferating transit-amplifying cell, which then subsequently adopt a mature endothelial cell phenotype connected to the circulation[10,13–15]. In particular, during skin wound healing, EVPs are found in the centre of the granulation tissue at day 1 (D1) and give rise to transit amplifying (TA) and differentiated endothelial (D) cells that can only be visualised from D3 post wounding[10]. The vessel formation peaks at D5 and then regresses giving room to a mesenchymal transition of ECs contributing to the scar[9]. Gene expression studies have identified two transcription factors (TFs), Sox9 and Rbpj, as being key markers distinguishing EVPs from other endothelial populations[14]. Sox9 has been reported in other stem cell compartments, in particular, in hair follicle stem cells but also in controlling chondrocytes lineage specification of mesenchymal cells as well as neuronal stem cells development[16–19]. Importantly, high expression of Sox9 has been implicated in driving/exacerbating fibrotic disease[20,21]. On the other hand, canonical Notch signalling through activity of its main effector transcription factor Rbpj is known to guide endothelial function and specification[22]. In other contexts, Rbpj and Sox9 have been shown to have antagonistic roles, whereby excess expression of Rbpj resulted in depleting Sox9 expression in chondrocytes driving chondrodysplasia[23]. Conversely, we have previously shown that when Rbpj is conditionally deleted in the endothelium, this resulted in accelerated EndMT, with excessive fibrosis and scar tissue formation observed in a skin excisional wound healing scenario[9].

These lines of evidence point towards a genetic interaction between Sox9 and Rbpj and prompted us to investigate a functional role of Sox9 specifically in the endothelium. We asked how these TFs orchestrate cell fate decision at the crossroad between vascular endothelium and mesenchymal transition. In this study we report that conditional deletion of Sox9 from the endothelium correlates with increased Rbpj expression and a significant reduction in EndMT and therefore fibrosis. Alternatively, indirect overexpression of Sox9 through loss of Rbpj or activation of hedgehog (HH) signalling in the endothelium produces opposite effects.

Using a pre-clinical model of wound healing, we establish the proof of principle that Sox9 gene knock-down with siRNA applied to wounds results in abrogation of EndMT and reduces the scar area, providing potential therapeutic avenues in blocking fibrosis.

## Results

**Expression of Sox9 in the mouse and human endothelium.** SOX9 protein expression has been associated with many stem cell types but hardly associated with the endothelium. In previous independent studies from our group, we have reported Sox9 mRNA expression through bulk or single cell RNA sequencing in murine EVPs isolated from either the aorta or the tumour endothelium[10,13]. Additionally, Sox9 mRNA expression has also been reported in coronary arteries and the aorta as observed in the Genotype-Tissue Expression (GTEx) project as well as in the Atlas of single cell RNA sequencing of the murine endothelium[24]. To explore and confirm the expression of the transcription factor SOX9 in the endothelium, a variety of scenarios were assessed. Thoracic and abdominal aortic tissue was isolated from wild-type C57Bl/6 mice before being processed for en face SOX9 immunofluorescence whole-mount staining. Stochastic clustering of SOX9 positive nuclei (labelled yellow) was observed throughout the aorta within cells harbouring the pan endothelial marker CD31(labelled red) (Fig. 1a). Similarly, in whole mounted normal skin, rare ECs expressed SOX9 (Fig. 1b). Additionally, we have tested the specificity of the SOX9 antibody and shown clear positive and specific staining of the nuclei of hair follicle bulge stem cells in skin (Supplementary Fig. 1a). Moreover, in a Sox9 knock-out situation no specific staining could be observed (see below). We also observed that the SOX9 staining is localised only to the endothelial layer and not in the underlying vascular smooth muscle layers (Supplementary Fig. 1b).

To better understand the frequency of SOX9 expression, aortic cells expressing VE-Cadherin and CD34 but devoid of hematopoietic markers (Lin-VE-Cad+CD34 + , Fig. 1c) were FACS sorted and separated into individual EVP (CD31$^{low}$ VEGFR2$^{low}$) and mature differentiated endothelial (D, CD31$^{Hi}$, VEGFR2$^{Hi}$) cells that were then cytospun onto slides to be stained for SOX9, ERG and RBPJ. As expected, only EVP cells stained positively for SOX9 as opposed to D cells. Importantly however, both EVP and D cells stained positively for the endothelial specific transcription factor marker ERG, demonstrating that SOX9 is indeed expressed in the endothelium (Fig. 1c). Moreover, RBPJ was found to be expressed in both EVP and D cells. EVPs that expressed RBPJ could be found to express SOX9 as well. Quantification of sorted aortic cells could show that only about a third of EVPs expressed SOX9, whereas most expressed RBPJ (Supplementary Fig. 1c and Fig. 1c). Finally, both RBPJ and SOX9 were significantly more frequently expressed at protein level in EVPs compared to D cells (SOX9 vs D cells $p < 0.001$; RBPJ vs D cells $p < 0.001$; Fig. 1c).

Previously, we have reported the presence of EVP cells in excisional wound granulation tissue at D1[10]. To confirm the expression of SOX9 in these cells, we next performed SOX9 immunostaining in wounds at this time-point[10]. Here we used a vascular lineage tracing mouse model, Cdh5-Cre$^{ER}$ RosaYFP, provided 10 days of tamoxifen via intraperitoneal injection (IP) to label the entire vasculature with YFP before full thickness excisional wounds were created on the back of mice. Co-labelling of the lineage tracing mark YFP and CD31 in a variety of tissues collected immediately after tamoxifen injection demonstrated a very strong overlap in signal demonstrating the specificity of endothelial lineage labelling upon Cre induction (Supplementary Fig. 2a). Upon tamoxifen induction and 24 h after wounding, most YFP + cells in the centre of the wound were individual EVP cells with SOX9 positive nuclei further

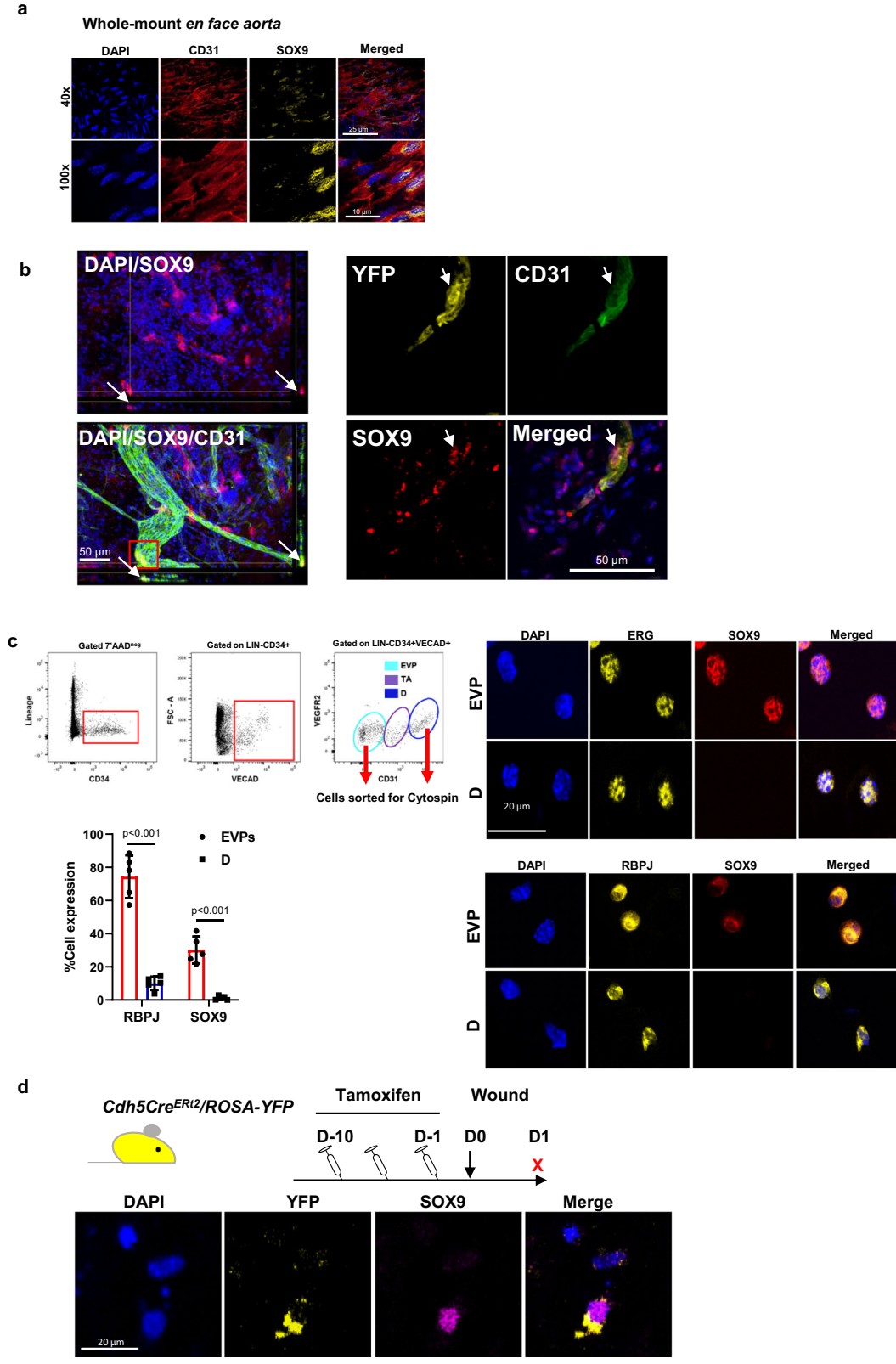

confirming the expression of SOX9 in YFP + ECs in a pathological scenario (Fig. 1d). Moreover, the use of a Sox9-Cre/ER reporter mouse showed Cre expression in the aortic and wound endothelium, demonstrating the expression of Sox9 at the RNA level in these tissues (Supplementary Fig. 2b).

These data in addition to our previous reports on mRNA expression provide strong evidence to suggest that the expression

of SOX9 is observed in ECs and is restricted to a specific EVP population within the endothelium.

**SOX9 expression is essential for EVP maintenance and quiescence.** To assess the molecular function of SOX9 in the endothelium we generated a *Sox9* endothelial specific conditional

**Fig. 1 Expression of Sox9 in the mouse and human endothelium. a** Whole-mount aorta staining demonstrating SOX9 nuclear staining among endothelial cells (Sox9 in yellow, CD31 in red). **b** Whole-mount skin staining demonstrating SOX9 expression in dermal vasculature (white arrow; SOX9 positive endothelial cells), immunofluorescent staining of skin sections collected from the *Cdh5-Cre^ER RosaYFP* mice displays nuclear SOX9 expression in a YFP + endothelial cell. **c** FACS sorting gating strategy, endovascular progenitor (EVPs) and mature differentiated endothelial (D) cells were FACS sorted and cytospun for staining. FACS sorted EVP showing positive co-staining for ERG, and SOX9 as well as RBPJ with SOX9. D cells are only positive for ERG and RBPJ but not SOX9. Quantification demonstrates a significantly larger percentage of EVPs are positive for RBPJ and SOX9 compared to D cells (***$p <$ 0.005; $n = 3$; cell sorted from three biologically independent animals; mean ± SD; $p$ value was calculated by two-way ANOVA with multiple comparison of row mean) **d** Immunofluorescence staining of day 1 (D1) wound sections collected from *Cdh5-Cre^ER RosaYFP* mice show that YFP + yellow cells within the centre are SOX9 positive.

knock-out that we termed a *Sox9^eKO (Sox9^fl/fl/Cdh5-Cre^ER RosaYFP)*. Both *Sox9^eWT (Sox9^+/+/Cdh5-Cre^ER RosaYFP)* and *Sox9^eKO* were provided tamoxifen injection IP for 10 consecutive days at adult age (10–12 weeks of age). The mice were culled, and the aortic tissue harvested for assessment of the endothelium (Fig. 2a). Upon tamoxifen injection, *Sox9* mRNA could not be detected in *Sox9^eKO* ECs. Similarly, SOX9 immunofluorescence failed to detect any remaining protein in the aorta from mice with conditionally deleted *Sox9* (Supplementary Fig. 3a). Fluorescence minus one (FMO) was used to determine negative and positive cut-offs for flow cytometry analysis (Supplementary Fig. 2b). *Sox9^eWT* aortas displayed the endothelial hierarchy previously described[10] among Lin-YFP + CD34 + cells. All three populations (EVP, TA and D) were observable (Fig. 2b). However, in the *Sox9^eKO* mice, a greater than two-fold significant depletion in the number of EVP cells was observed. This coincided with a significant increase in the D population numbers (**$p < 0.01$; Fig. 2c). Of importance, cell cycle regulators such as *Il33, p16, p21* and *p57* showed over fourfold reduction in mRNA expression in EVPs isolated from *Sox9^eKO* mice suggesting a role for Sox9 in maintaining EVPs in quiescence (***$p < 0.001$; Fig. 2d)[25]. We next addressed the functional importance of SOX9 in the endothelium in disease.

**Loss of Sox9 in the endothelium significantly impairs EndMT and reduces scarring during wound healing.** Given the importance of EVPs in neo-vessel formation during wound healing, we used excisional wounds to evaluate the functional importance of SOX9 expression in EVPs. Here *Sox9^eWT* and *Sox9^eKO* mice were injected IP for 10 days with tamoxifen before undergoing full thickness large skin excisional (1.5 cm × 1.5 cm) wounds on their dorsal skin allowing us to evaluate wound healing and scar tissue formation (Fig. 3a). Large excisional wounds were chosen to allow a direct macroscopic measurement of the scar area and results were confirmed in small 6 mm excisional wounds through microscopic examination. The wounds from each group were monitored every 3 days and wound area was measured (Fig. 3b, c; $n = 8$ per group). Intriguingly, the *Sox9^eKO* mice displayed significantly faster wound healing capacity compared to the *Sox9^eWT* controls from D12 onwards (*$p < 0.05$; **$p < 0.01$ vs *Sox9^eWT*).

The scar quality was then assessed in 6 mm full thickness punch biopsy wounds on the dorsum of *Sox9^eWT* and *Sox9^eKO* mice. We had observations of reduced scarring using Sirius red staining upon *Sox9* deletion in the endothelium (Fig. 3e). Wounds were also harvested at day 5 (D5) and 7 (D7) to evaluate EndMT (Supplementary Figs. 4, 5 and Fig. 4a). We first assessed if changes in the endothelial fate could be observed via flow cytometry. We established a new gating strategy to allow tracking of all populations labelled with YFP upon lineage tracing by gating on live Lin-YFP + cells (Supplementary Fig. 4). Gated cells were then distinguished based on their level of CD34 and CD31 surface expression. Comparison between the two gating strategies allowed the identification of EVP and D cells as well as

CD31 + cells that did not express CD34 extending from the D population. This new gating strategy allowed the visualisation of a CD34-CD31-population that expressed YFP. Further staining showed that the latter population expressed mesenchymal markers such as CD26 or PDGFRα at higher levels compared to endothelial populations. Moreover, upon sorting and immunostaining, these cells expressed αSMA and not CD31, in contrast to D cells (Supplementary Fig. 4). This population was therefore labelled mesenchymal (M) and could be identified in the aorta but at relatively higher levels in the normal skin.

Using this new gating strategy, we next performed flow cytometry evaluation of skin wounds at D5 and D7 (Supplementary Fig. 5a, b). As reported previously by us and others, the total number of ECs (Lin- and VE-cadherin+ cells) reduced between D5 and D7 reflecting the peak in angiogenesis often reported in D5. *Sox9^eKO* animals had equal number of ECs in D5 but higher numbers in D7 compared to controls. We observed a significant reduction in the number of EVP cells within the granulation tissue of wounds at D5 in *Sox9^eKO* in comparison to controls ($p < 0.001$ vs *Sox9^eWT*; Supplementary Fig. 5b) although this difference was not significant in D7 given the previously reported reduction of EVP cells in wounds at D7[10]. This resulted in a relative increase in D cells but no significant change in the absolute number of D cells at both D5 and D7. Next, using the new gating strategy, we could evaluate EndMT through quantification of the M population. The M population was deemed to exemplify EndMT as it originated from ECs given its YFP lineage label and had subsequently lost CD34 and CD31 and gained mesenchymal markers such as CD26[26] or PDGFRα. Although M cells did not differ in numbers at D5, they were significantly reduced at D7, a time-point past peak angiogenesis when EndMT starts to occur as we have reported previously[9] ($p = 0.024$ vs *Sox9^eWT*, Supplementary Fig. 5b).

Next we evaluated EndMT in tissue sections using three classical mesenchymal markers, α-SMA, FSP1 and SLUG, alongside endothelial markers YFP and CD31 at D7[9](Fig. 4b). The number of vessels and individual cells that co-expressed YFP + α-SMA + CD31 + was significantly reduced in *Sox9^eKO* mice compared to controls ($p < 0.001$ vs *Sox9^eWT* Fig. 4b, c). In this assessment, we took extreme care to avoid counting mature vessels harbouring α-SMA + pericyte on their periphery and only true overlap of the triple staining was taken into account. Additionally, the number of vessels that had entirely lost endothelial phenotype (YFP + α-SMA + CD31-) was also significantly reduced in *Sox9^eKO* mice compared to controls ($p < 0.001$ vs *Sox9^eWT* Fig. 4c). Of importance, the latter category included individual dermal cells detached from the structure of classical blood vessels (Supplementary Fig. 5c). We next assessed a key EndMT transcription factor SLUG[27,28] as well as the fibroblast marker FSP1 (S100A4) expression and identified again a significant reduction in both vessels and cell counts in YFP + structures that had lost CD31 expression ($p < 0.001$ vs *Sox9^eWT*; Fig. 4c). Overall, these findings suggest that the deletion of *Sox9* specifically from the endothelium reduces the likelihood of

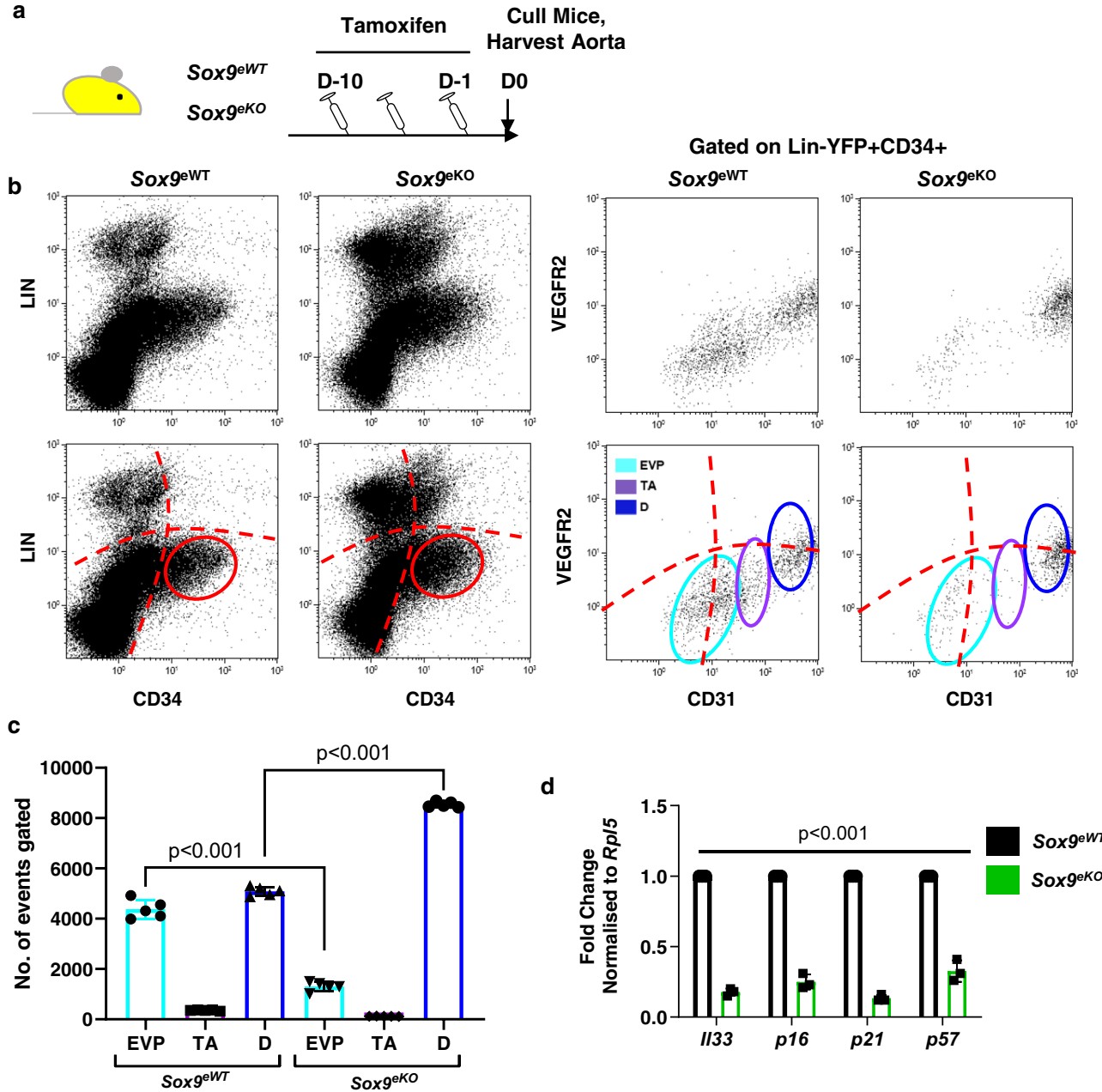

**Fig. 2 SOX9 expression is essential for endovascular progenitor clonogenicity and quiescence. a** Schematic diagram demonstrating experimental set up using conditional vascular specific *Sox9* knockout mice (*Sox9fl/fl/Cdh5-CreER RosaYFP – Sox9eKO*) and sample collection 12 weeks post-induction. **b** Flow cytometry plots showing the aortic endothelium harbour a distinct CD34 positive, lineage (Lin) negative population (red gate) that is entirely YFP positive, as determine by fluorescence minus one (FMO) analysis. Three distinct populations were observed based on CD31 and VEGFR2 expression (from left to right endovascular progenitor EVP; transit amplifying TA; differentiated D) showing the endothelial hierarchy and changes between wild-type controls (*Sox9eWT*) and *Sox9eKO* mice after 10 days tamoxifen. **c** Number of EVP is significantly reduced in *Sox9eKO* ($p < 0.001$ vs *Sox9eWT*; $n = 3$ biologically independent animals; mean ± SD). **d** qPCR analysis of genes associated with endothelial quiescence in EVP (***$p < 0.001$ vs *Sox9eWT*; $n = 3$; mean ± SD).

EndMT, thus resulting in reduced fibrosis and scarring in skin wounds.

**Sox9 deletion results in increased Notch signalling with reduced expression of TGFβ and EndMT target genes**. To understand the gene expression consequences of SOX9 deletion in the endothelium, aortic tissue from *Sox9eWT* and *Sox9eKO* mice was harvested and EVP were FACS sorted and processed for analysis by quantitative polymerase chain reaction (qPCR). Key notch signalling effector, *Rbpj*, increased by over 4.5-fold in *Sox9eKO* compared to controls and was associated with an upregulation of

expression in canonical Notch signalling target genes such as *Hey1* and *Hes1*. Classical EndMT genes such as *TGF-β*, *Snail*, *Slug* and *Twist* were all downregulated by over 4-fold, suggesting the upstream importance of *Sox9* in this pathway classically attributed to TGFβ signalling[28,29] ($p < 0.002$ vs *Sox9eWT*; Fig. 4d, e). The changes in gene expression further support that loss of *Sox9* results in decrease in EndMT and its associated genes. It also resulted in an increase in *Rbpj* expression possibly driving the maintenance of an endothelial fate, blocking EndMT. Indeed, we previously reported that loss of Rbpj in the endothelium resulted in increased EndMT and scarring in the context of wound healing[9].

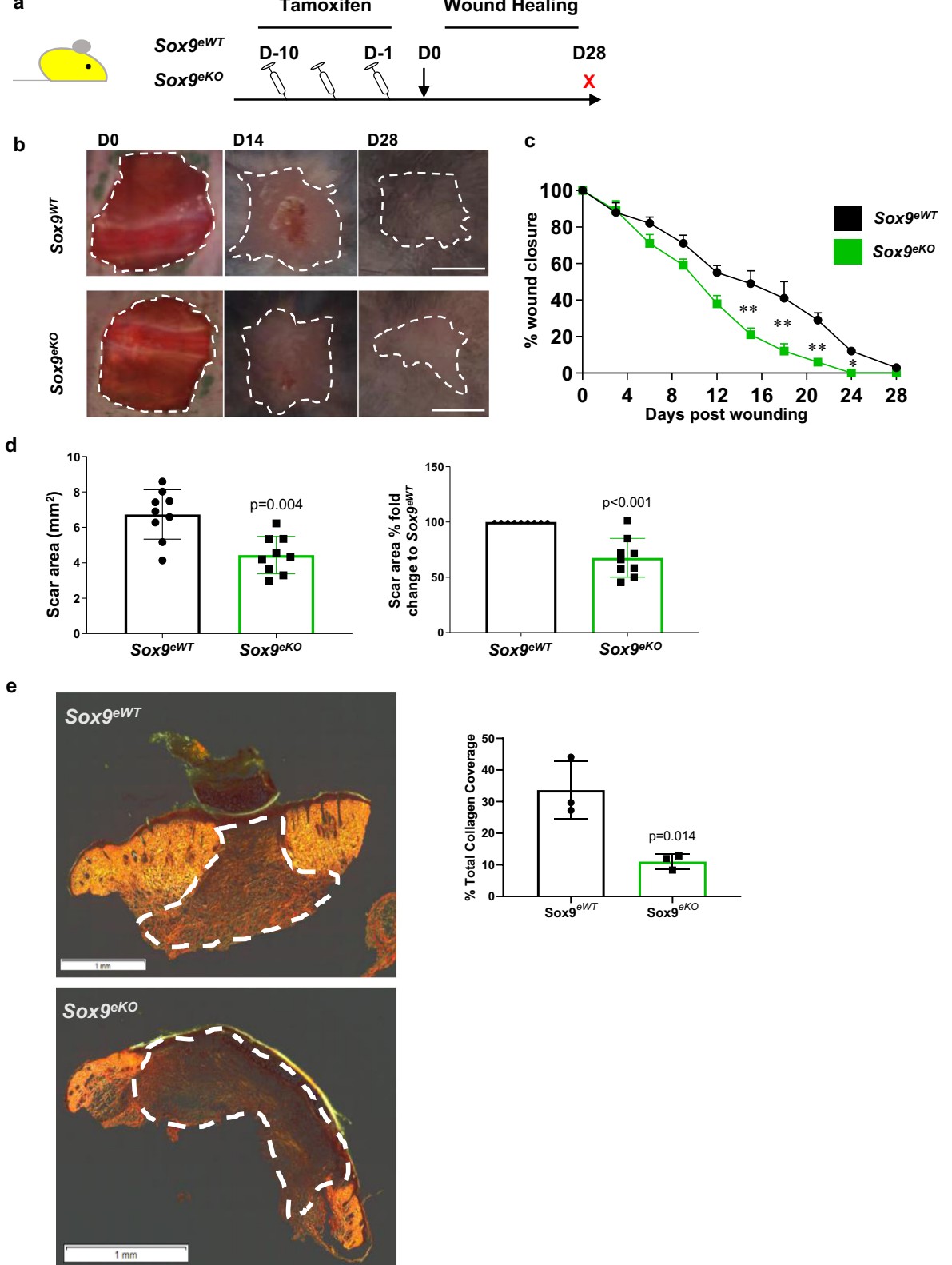

**Fig. 3 Loss of *Sox9* in the endothelium significantly impairs EndMT and reduces scarring during wound healing. a** Schematic diagram demonstrating experimental set up using conditional vascular specific *Sox9* knockout mice (*Sox9fl/fl/Cdh5-CreER RosaYFP – Sox9eKO*). **b** Digital photography images showing macroscopic wound healing over the course of 28 days (D28) post wounding between wild-type controls (*Sox9eWT*) and *Sox9eKO*. Scale bar = 5 mm. **c** Skin wound healing rate as a percentage of initial wounded area was significantly increased in *Sox9eKO* compared to *Sox9eWT* (*$p < 0.05$; **$p < 0.01$ vs *Sox9eWT*; $n = 8$ biologically independent animals; mean ± SD; *p* value was calculated by two-tailed unpaired *t* test). **d** After 28 days, a significant decrease in macroscopic scar area was observed in *Sox9eKO* mice compared to *Sox9eWT* ($p = 0.004$; $n = 8$ biologically independent animals; mean ± SD; *p* value was calculated by two-tailed unpaired *t* test) **e** Sirius red collagen staining of both 6 mm and large wounds demonstrates a significant decrease in total collage coverage within the granulation tissue in *Sox9eKO* compared to *Sox9eWT* ($p = 0.0014$ vs *Sox9eWT*; $n = 3$ biologically independent animals; mean ± SD; *p* value was calculated by two-tailed unpaired *t* test).

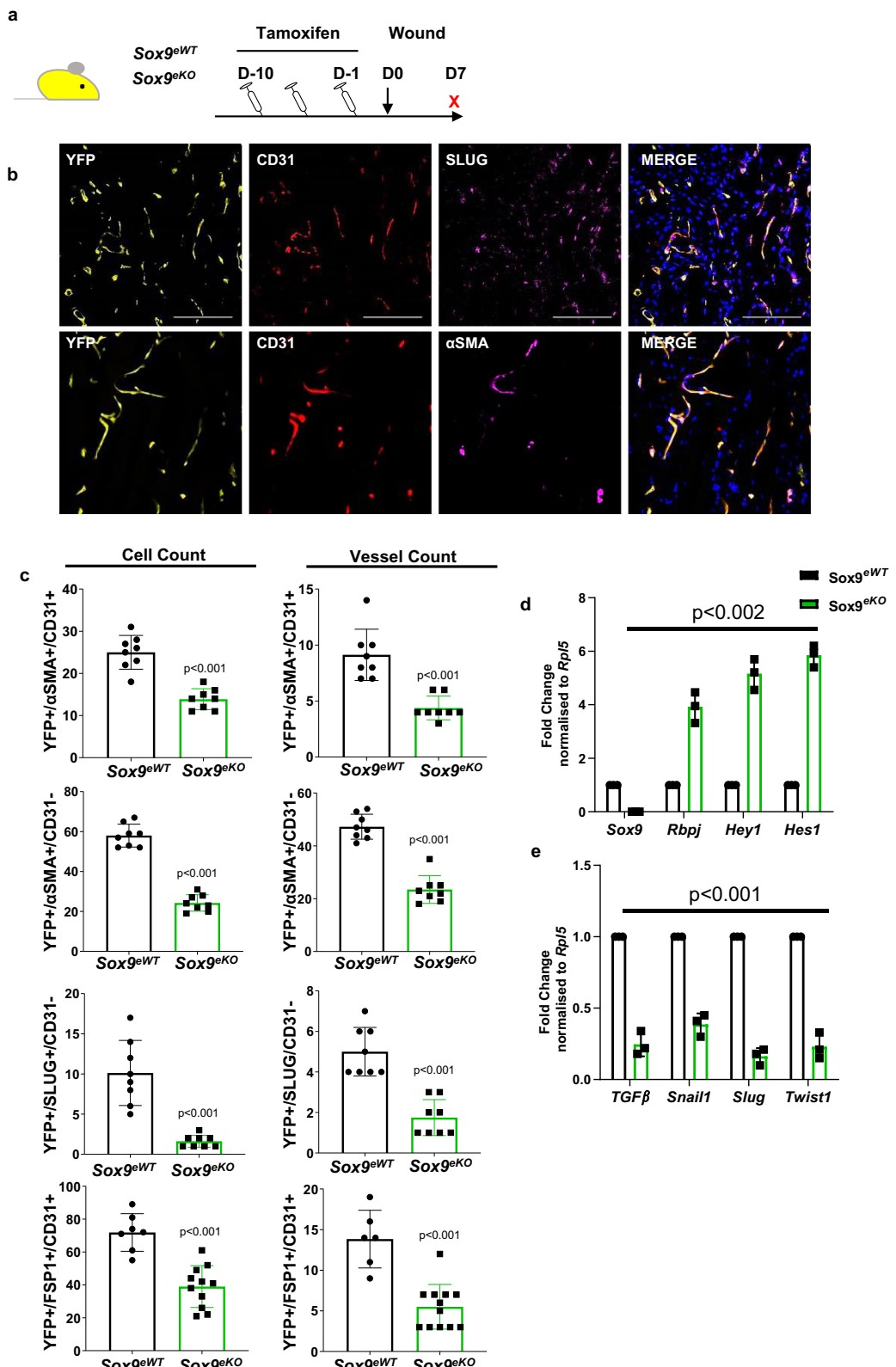

**Fig. 4 Quantification of endothelial to mesenchymal transition (EndMT) upon loss of SOX9. a** Schematic diagram demonstrating experimental set up using conditional vascular specific *Sox9* knockout mice (*Sox9^fl/fl^/Cdh5-Cre^ER^ RosaYFP – Sox9^eKO^*). **b** Examples of immunofluorescence staining of skin wound sections at day 7 (D7) with endothelial markers CD31, lineage marker YFP and EndMT markers SLUG or αSMA. Scale bar = 50 μm. **c** Quantification of EndMT showing overlap of YFP + cells with either endothelial and mesenchymal markers such as SLUG, αSMA and FSP1 (**p < 0.01 vs *Sox9^eWT^*; n = 5 biologically independent animals; mean ± SD; p value was calculated by two-tailed unpaired t test). **d**, **e** qPCR analysis of key endothelial and EndMT genes in EVP from *Sox9^eKO^* compared to *Sox9^eWT^* (**p < 0.01; ***p < 0.001 vs *Sox9^eWT^*; n = 3; cells were sorted from three groups of five biologically independent animals; mean ± SD; p value was calculated by two-way ANOVA with multiple comparison of row mean).

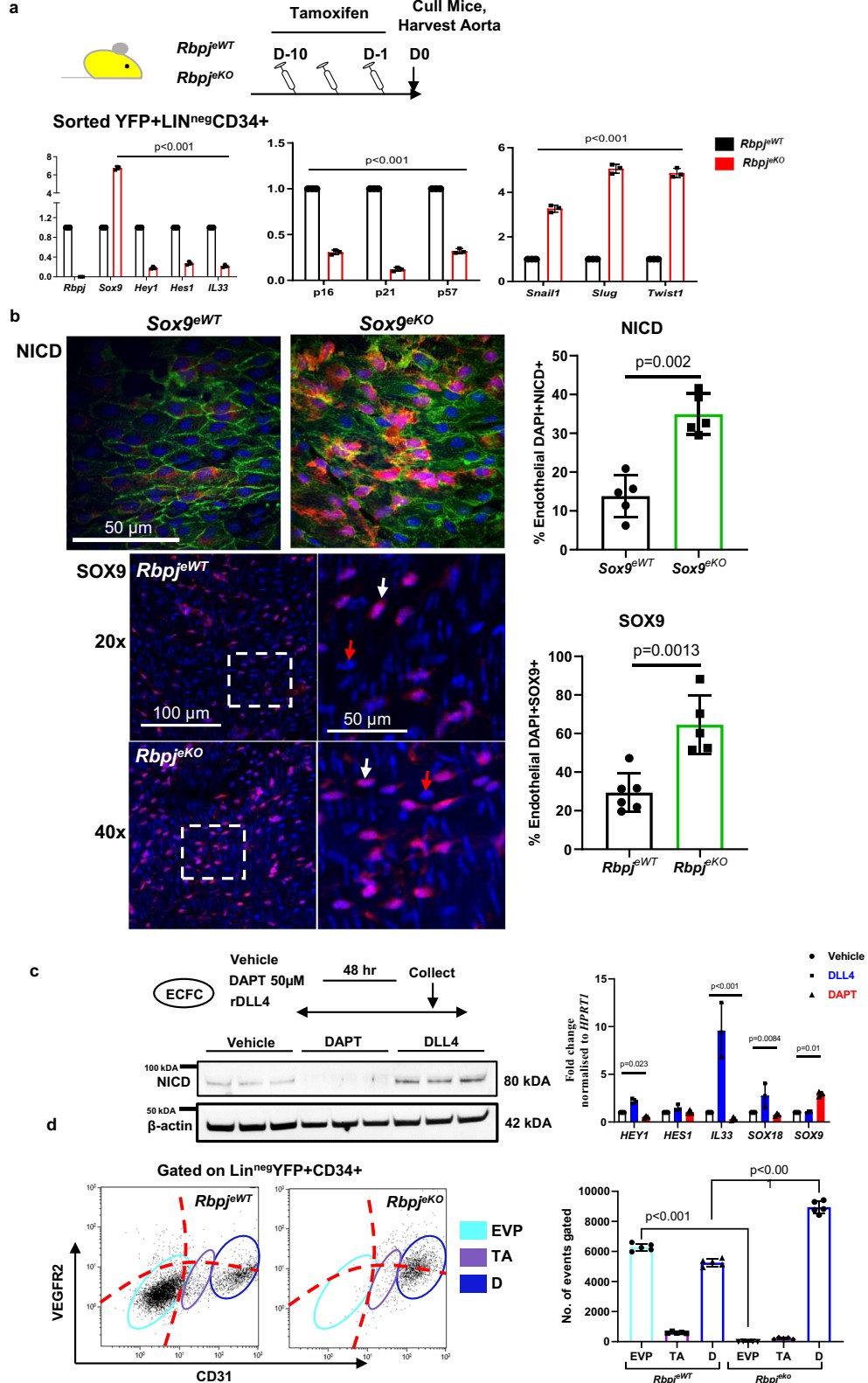

We therefore sought to identify if a reciprocal interplay could be observed between SOX9 and RBPJ that specifically mark the EVP population given the opposite phenotypes observed upon conditional deletion. Therefore, we used an endothelial specific knockout of *Rbpj^eKO* (*Rbpj^fl/fl*/*Cdh5-Cre^ER* *RosaYFP*). Here, following 10 days of tamoxifen injections, aortic tissue was harvested and evaluated under flow cytometry and qPCR (Fig. 5a).

qPCR demonstrated the abrogation of *Rbpj* expression as well as canonical Notch signalling, since *Hes1* and *Hey1* were markedly reduced in the entire endothelium (***$p < 0.001$; Fig. 5a). Following *Rbpj^eKO*, the expression of *Sox9* also increased by over 7-fold compared to controls. Moreover, *Rbpj^eKO* displayed a significant increase in EndMT genes *Snail*, *Slug* and *Twist* (**$p < 0.01$; ***$p < 0.001$; Fig. 5A). Genes involved in endothelial

**Fig. 5 Co-inhibitory interplay of SOX9 and Notch signalling in the endothelium. a** Schematic diagram demonstrating experimental set up using conditional vascular specific *Rbpj* knockout mice (*Rbpj*$^{fl/fl}$/Cdh5-Cre$^{ER}$ RosaYFP – *Rbpj*$^{eKO}$). qPCR analysis of key endothelial, quiescence and EndMT genes in the endothelial fraction (YFP + Lin$^{neg}$CD34 + ) (**$p < 0.01$; ***$p < 0.001$ vs *Rbpj*$^{eWT}$; $n = 3$; cells were sorted from three groups of five biologically independent animals; mean ± SD; $p$ value was calculated by two-way ANOVA with multiple comparison of row mean). **b** Aortic whole-mount en face staining for activated-NOTCH1 intracellular domain (NICD) demonstrates significant increase in the nuclear localisation of activated-NICD in the endothelium after conditional deletion of SOX9. The deletion of RBPJ resulted in a significant increase in the number of SOX9 positive endothelial cells. (**$p < 0.01$, ***$p < 0.001$ vs e*WT*; $n = 4$ aorta from four biological independent animals; mean ± SD; $p$ value was calculated by two-tailed unpaired $t$ test). **c** Western blot analysis of human endothelial colony forming cells, ECFCs shows DAPT inhibits activated-NICD accumulation, whereby rDLL4 treatment significantly upregulates activated-NICD expression showing the possibility to modulate canonical Notch signalling in vitro in ECFCs. qPCR analysis of key NOTCH pathway genes and SOX9 after modulation of canonical Notch pathway (*$p < 0.05$ **$p < 0.01$; ***$p < 0.001$ vs Vehicle; $n = 3$; cells were isolated from three biologically independent donors; mean ± SD; $p$ value was calculated by two-way ANOVA with multiple comparison of row mean) **d** Using flow cytometry three distinct populations were observed based on CD31 and VEGFR2 expression (from left to right endovascular progenitor EVP; transit amplifying TA; definitive differentiated D) showing the endothelial hierarchy and changes between wild-type controls (*Rbpj*$^{eWT}$) and *Rbpj*$^{eKO}$ mice after 10 days tamoxifen. Number of EVP is significantly reduced in *Rbpj*$^{eKO}$ ($p < 0.001$ vs *Rbpj*$^{eWT}$; $n = 3$ biologically independent animals; mean ± SD; $p$ value was calculated by two-way ANOVA with multiple comparison of row mean).

quiescence such as *Il33*, *p16*, *p21* and *p57* were also significantly downregulated (***$p < 0.001$). We further explored this opposite regulation of Sox9 and RBPJ/Notch signalling (Fig. 5b). In *Sox9*$^{eKO}$ aortas, there was a significant increase in nuclear NOTCH intracellular domain (NICD) reflecting the activation of canonical NOTCH signalling. Indeed over 30% of nuclei displayed nuclear NICD in *Sox9*$^{eKO}$ ($p = 0.002$ vs *Sox9*$^{eWT}$; Fig. 5b). In contrast, *Rbpj*$^{eKO}$ aortas displayed significantly higher levels of SOX9 expression as compared to controls ($p = 0.0013$ vs *Rbpj*$^{eWT}$; Fig. 5b). These findings corroborate at the protein level and in vivo, with the results obtained through qPCR to suggest that RBPJ negatively regulates SOX9 expression and that SOX9 negatively regulates canonical NOTCH signalling as well as RBPJ itself. We further tested this relationship in vitro. Human endothelial colony forming cells (ECFCs) were exposed to either vehicle control or to either the NOTCH ligand DLL4 coated plates or to the canonical NOTCH pathway gamma-secretase inhibitor DAPT (Fig. 5c). Each intervention had the expected outcome in terms of NOTCH pathway activation, NICD accumulation and affected gene expression of downstream effectors such as *HEY1*, or other genes important for endothelial function such as *IL33* or *SOX18*. This system allowed us to explore the effect of canonical NOTCH signalling on SOX9 expression. DAPT treatment resulted in a significant increase in SOX9 ($p = 0.01$ vs Vehicle; Fig. 5c) supporting the claim that active NOTCH signalling and RBPJ negatively regulate *Sox9* expression. However, DLL4 treatment and NOTCH pathway activation did not further affect SOX9 expression.

Similar to *Sox9*$^{eKO}$, a depletion in the EVP population in the aorta of *Rbpj*$^{eKO}$ compared to *Rbpj*$^{eWT}$ controls was observed under flow cytometry (***$p < 0.001$; Fig. 5d). Similarly, in wounds at D5, conditional ablation of *Rbpj*$^{eKO}$ resulted in a reduction in EVPs. However, unlike *Sox9*$^{eKO}$ animals, in *Rbpj*$^{eKO}$ the number of D cells in the granulation tissue was significantly reduced (***$p < 0.001$; Supplementary Fig. 6b). Additionally, amongst Lin-YFP + , there was a significant increase in M (CD34-CD31-) cells in *Rbpj*$^{eKO}$ compared to controls, in line with the increased EndMT and fibrosis previously reported in these mice ($p = 0.02$ vs *Rbpj*$^{eWT}$; Supplementary Fig. 6c)[9].

These findings suggest that *Sox9* and *Rbpj* genetically interact via a negative feedback loop and drive opposite EndMT outcomes upon expression in the endothelium. Of significant interest, in the endothelial hierarchy described in both murine[10] and human ECs[9], EVPs in mice and meso-endothelial progenitors in human placenta have higher levels of expression of *Sox9* whereas fully differentiated D cells or ECFCs have high Notch signalling. This duality brought us to consider EVPs as a progenitor prompt to a fate choice towards either endothelium through increased Notch

signalling (TA and D cells) or towards EndMT through increased *Sox9* activity.

**Activation of HH signalling drives EndMT**. HH signalling has been implicated as a key regulator of *Sox9* expression and function driven by GLI1 translocation into the nucleus. Additionally, previously published RNA sequencing data from our lab showed that *Gli1* is significantly upregulated only in the EVP population of the murine endothelium[10,13]. In a different context, HH signalling has also been shown to reduce Notch signalling. To assess the molecular function of HH signalling in the endothelium we generated a *Ptch1* endothelial specific knock-out that we termed a *Ptch1*$^{eKO}$ (*Ptch1*$^{fl/fl}$/Cdh5-Cre$^{ER}$ RosaYFP) to specifically drive HH signalling gain of function in the endothelium. Both *Ptch1*$^{eWT}$ and *Ptch1*$^{eKO}$ were provided tamoxifen injections and the aortic tissue was harvested for assessment of the endothelium (Supplementary Fig. 7a). To firstly assess if HH signalling was activated, YFP + EVP and D cells were FACS sorted from *Ptch1*$^{eKO}$ for qPCR analysis. *Ptch1* was significantly reduced, whereas *Smo*, *Gli1* and *Sox9* were all significantly upregulated in the EVPs in comparison to D in the *Ptch1*$^{eKO}$ ($p < 0.001$ vs D cells *Ptch1*$^{eKO}$; Supplementary Fig. 7b). The same was also observed when comparing EVP from *Ptch1*$^{eKO}$ to EVP from *Ptch1*$^{eWT}$ ($p < 0.001$ EVP *Ptch1*$^{eWT}$ vs EVP *Ptch1*$^{eWT}$ Supplementary Fig. 7b). We then analysed the aortic endothelium under flow cytometry. The *Ptch1*$^{eKO}$ mice almost half (48%) of all Lin-YFP + cells had lost CD34 expression, compared to only 13% for *Ptch1*$^{eWT}$ suggesting an accelerated loss of endothelial phenotype (Supplementary Fig. 7c). To understand differences in phenotypes between the different conditional knockout models, *Ptch1*$^{eKO}$, *Sox9*$^{eKO}$ and *Rbpj1*$^{eKO}$ aortas were compared to controls (Cdh5-Cre$^{ER}$ RosaYFP). Despite slight differences, 4 weeks after tamoxifen injection, there was no significant changes in aortic endothelial numbers (as a proportion of live cells, Fig. 6b). All three conditional gene deletions resulted in strong reductions in EVP numbers, suggesting they affected progenitor function. However, they affected differently the other endothelial populations. The most dramatic changes occurred upon loss of *rbpj* resulting in increases in D and M cells. Loss of *ptch1* also resulted in increased M cells, whereas *sox9* loss resulted in a slight increase in D cells but did not significantly affect the endothelial cell phenotype distribution in homoeostasis. Overall, this showed that despite similar effects on EVPs, these different genes regulate endothelial cell fate slightly differentially during homoeostasis (Fig. 6C).

This also suggested that upon HH activation both D cells and M cells could be obtained to explore the duality between Sox9 and Notch signalling. We therefore FACS sorted Lin-YFP + CD34 + (to reflect mostly D cells, given the major reduction in EVPs) and

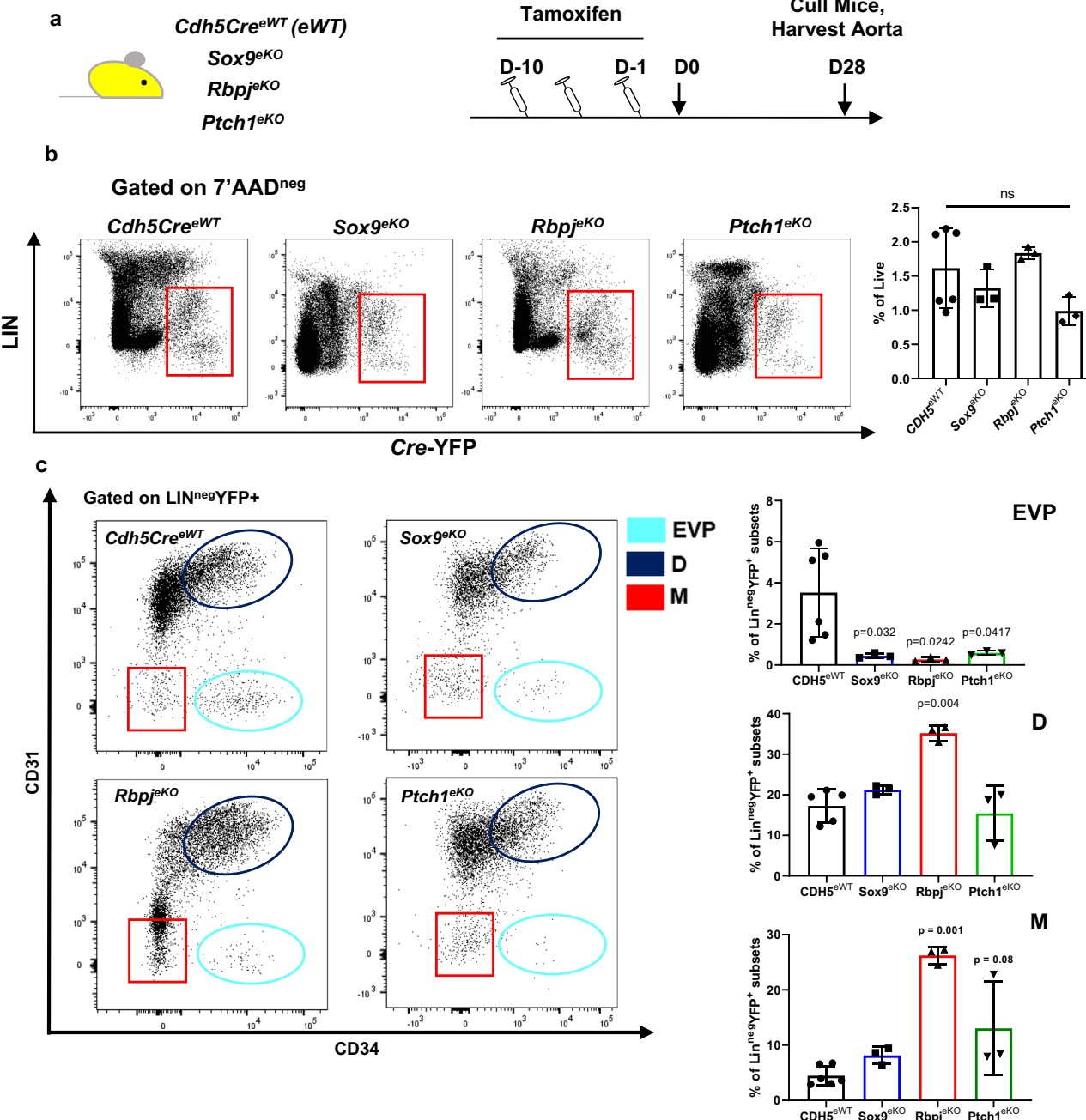

**Fig. 6 Conditional deletion of key factors in the endothelium drives aortic progenitor directional fate-choice differentiation. a** Schematic diagram demonstrating experimental set up of conditional *Sox9eKO, RbpjeKO or Ptch1eKO* knockout mice. **b** Flow cytometry plots showing the total endothelial population (YFP + Lin^neg) within the aorta was not affected by the conditional deletion of *Sox9, Rbpj* or *Ptch1* (ns *p* > 0.05 vs *eWT*; *n* = 3 biologically independent animals; mean ± SD; *p* value was calculated by one-way ANOVA with multiple comparison to *eWT*). **c** Subsequent populations were then demarcated by CD31 and CD34 expression, demonstrating the significant depletion of EVPs in all the knockout mice compared to wildtype with different fates adopted by endothelial cells such as increases within the M population after the deletion of *Rbpj* and *Ptch1* (**p* < 0.05, ***p* < 0.01 vs *eWT*; *n* = 3 biologically independent animals; mean ± SD; *p* value was calculated by one-way ANOVA with multiple comparison to *eWT*).

Lin-YFP + CD34− (to mostly reflect mesenchymal M cells) cells for analysis by qPCR. Notch signalling target genes (*Rbpj*, *Hey1*, *Hes1*) and endothelial gene *Pecam1* were significantly increased in expression in the CD34 + 'endothelial' group and EndMT genes showed only slight reduction. However, in the CD34− 'mesenchymal' group, Notch family gene expression and *Pecam* was completely abrogated by over fourfold reduction in expression, however *Sox9* expression was significantly increased along with the expression of downstream EndMT genes (*Snail, Slug, Twist1, Twist2*) (*p* < 0.005 vs *Ptch1eWT*; Supplementary Fig. 7d). These

data further show the importance of the opposing role of SOX9 and Notch signalling in two different endothelial fates and highlight the role of HH signalling in this fate decision.

**Increased HH signalling in the endothelium results in scarring and increased EndMT in skin wounds.** To evaluate whether an increase in HH signalling results in accelerated EndMT and thus exacerbating fibrosis, as expected through the increase of SOX9 and decrease of Notch signalling, we again used the full thickness

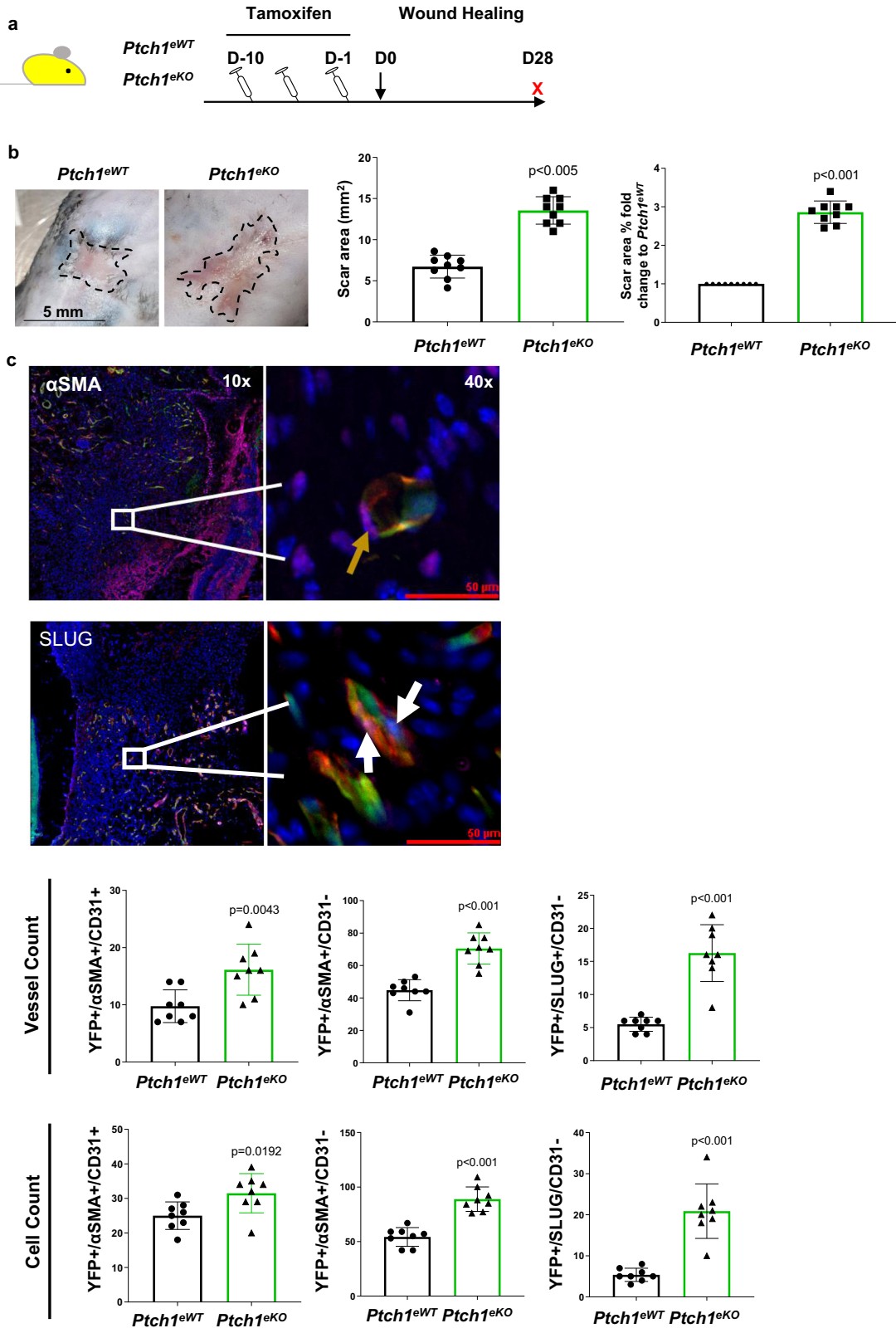

skin excisional (1.5 cm × 1.5 cm) wounds on the dorsal flanks on the mice. Prior to wounding both *Ptch1^eWT* and *Ptch1^eKO* were provided tamoxifen and wounds were collected at day 28 (D28) (Fig. 7a). At D28, macroscopic observation and quantification demonstrated the scar areas were over threefold larger in the *Ptch1^eKO* compared to controls ($p < 0.005$ vs *Ptch1^eWT*; Fig. 7b). We then conducted 6 mm full thickness punch biopsy wounds on

the dorsal flanks of the *Ptch1^eWT* and *Ptch1^eKO* mice following 10 days of IP tamoxifen injections. Wounds were then harvested at day 7 (D7) and assessed for EndMT using SLUG and α-SMA. As expected, a significant increase in YFP + vessel and cell counts expressing both SLUG and α-SMA were observed in *Ptch1^eKO* wounds compared to controls. Importantly, this also coincided with a significant increase in YFP + αSMA + ($p = 0.0043$ vs

**Fig. 7 Increased Hedgehog signalling in the endothelium results in scarring and increased endothelial to mesenchymal transition (EndMT) in skin wounds. a** Schematic diagram demonstrating experimental set up using conditional vascular specific $Ptch1^{eKO}$ knockout mice ($Ptch1^{fl/fl}/Cdh5$-$Cre^{ER}$ RosaYFP – $Ptch1^{eKO}$). **b** Photo images and scar surface assessment showing increase scar area at day 28 (D28) post wounding between wild-type controls ($Ptch1^{eWT}$) and $Ptch1^{eKO}$ (**$p < 0.01$ vs $Ptch1^{eWT}$; $n = 8$ biologically independent animals; mean ± SD; $p$ value was calculated by two-tailed unpaired $t$ test). **c** Quantification of skin wound sections at day 7 (D7) with endothelial markers CD31 and YFP and EndMT markers SLUG and αSMA. High magnification representative images show YFP + (green) nuclei co-expressing αSMA (purple) and CD31 (red) marked by white arrow. Additionally, YFP + nuclei exhibit nuclear expression of SLUG (yellow arrow) (**$p < 0.01$; ***$p < 0.001$ vs $Ptch1^{eWT}$; $n = 8$ biologically independent animals; mean ± SD; $p$ value was calculated by two-tailed unpaired $t$ test).

$Ptch1^{eWT}$; Fig. 7c) and YFP + SLUG + vessels and single cells ($p < 0.001$ vs $Ptch1^{eWT}$; Fig. 7c) entirely losing CD31 expression.

**Reduced scarring upon use of siRNA to inhibit Sox9 in wound healing.** siRNA as a topical agent in treating wounds to improve healing and/or to reduce scarring have been used previously in clinical trials[30]. To evaluate the potential benefit of targeting Sox9 to reduce EndMT in wounds, we delivered siRNA against $Sox9$ using a topically applied pluronic gel directly onto full thickness skin excisional ($1.5\,cm \times 1.5\,cm$) wounds on the dorsal flanks of wild-type (C57Bl/6 J) mice. The gel was applied twice daily between days 5–14 (D5–D14) and wounds harvested at D28 (Fig. 8a). The wounds and mice were closely monitored, with images taken every few days. Mice treated with $Sox9$ siRNA displayed enhanced wound healing and accelerated closure by D5, mirroring our observations in the $Sox9^{eKO}$ used earlier (Fig. 8b; $n = 8$ per group). Additionally, when scar area was assessed there was a 38% reduction following $Sox9$ siRNA treatment compared to scrambled controls (Scr) ($p < 0.001$ vs Scr; Fig. 8c). EndMT was then quantified using 6 mm full thickness punch biopsy wounds on the dorsal flanks of mice. Mice were treated with topical pluronic gel twice daily from D1–6 with wounds harvested on D7. There was a significant reduction in CD31 + vessels and cells expressing either SLUG or α-SMA in mice treated with $Sox9$ siRNA ($p < 0.001$ vs Scr; Fig. 8d). Overall, the results of this study support the concept for blocking $Sox9$ expression in the vasculature during wound healing to block EndMT and reduce scar area.

## Discussion

The profound adaptive and regenerative capacity of the adult endothelium are attributed to a population of stem or progenitor cells that resides within the vasculature throughout all organ systems[10,11,31,32]. However, a functional and molecular definition of vascular endothelial progenitors remains in infancy. The description of $Sox9$ and $Rbpj$ as important TFs in demarcating EVP has potentiated more precise investigation into the mechanisms underlying vascular endothelial regeneration in homoeostatic turnover and pathological EndMT[9,10]. Specifically, this project employed functional fate-mapping in vivo and gene expression analyses to determine the effect of EVP fate choice during vascular homoeostasis and pathology. Here we show that modulation of $Sox9$ expression in the endothelium through its conditional knockout or its upregulation via reduced Notch signalling or increased HH signalling affects EVP fate choice between an endothelial or a mesenchymal phenotype. Increased expression of $Sox9$ was associated with increased EndMT and fibrosis whereas Sox9 conditional deletion substantially reduced EndMT and scarring (Fig. 9).

A major finding in the present study was the molecular modulation of endothelial fate. As the key TFs modulating this fate are expressed at significantly higher levels in EVPs as compared to other endothelial populations, one can assume that mesenchymal transition as well as endothelial differentiation

emanates from the progenitor EVP cells rather than D cells that do not express $Sox9$. We and others have described the existence of bipotent endothelial progenitors in the human term placenta able to give rise to both endothelial and mesenchymal colonies in vitro[9]. Of interest, the transcriptional profile as well as the surface expression of these cells is highly reminiscent of EVPs. Similarly, others have reported bipotent endothelial progenitors in mice[33] or upon derivation from iPSCs[34,35]. Our study is limited by the inability to directly trace the fate of EVPs and D cells independently to demonstrate their potency. However, given the gene expression profile of Sox9, our findings suggest that EVPs have the capacity to evolve towards an endothelial phenotype and contribute to blood vessel formation or to enter mesenchymal transition towards myofibroblasts and contribute to scarring.

Trans-differentiation of endothelial to mesenchymal cells through EndMT has recently been associated with fibrotic disease in diverse cellular contexts, including the exacerbation of cardiac, pulmonary fibrosis and atherosclerotic plaque instability[3,29]. Using a variety of lineage tracing models and pro-fibrotic markers, ECs were identified to contribute to fibrosis by transitioning into pro-fibrotic mesenchymal cells, specifically fibroblasts and myofibroblasts[3,9,10]. Therefore, an understanding of the molecular basis of EndMT in diverse niches may clarify the clinical impact of EndMT and identify therapeutic targets. Although not previously associated with EndMT, $Sox9$ has been attributed to the abnormal ECM remodelling in the kidney, cardiac and liver fibrosis[3,20,21]. In kidney fibrosis, TGF-β, a known major regulator of the EndMT pathway, was shown to upregulate $Sox9$ expression exacerbating tissue fibrosis and leading to further kidney disease[3] whereas here we show that $Sox9$ deletion also affected TGFβ downstream targets. $Sox9$ is also vital for mesenchymal stem cell maintenance, acting as a key regulator in differentiation into adipocyte, chondrocyte and osteocyte lineages[36,37]. As $Sox9$ is described to regulate both stem cell maintenance and mesenchymal differentiation, its upregulation within EndMT suggests a dual role for $Sox9$ to maintain both EVP stemness and a mesenchymal fate plasticity. Our study demonstrated the importance for $Sox9$ in maintaining EVP in a quiescent state as well as maintaining colony-formation capacity. Its loss resulted in rapid transition from a progenitor to D fate but more importantly a significant reduction in scar tissue in the skin wound healing scenario. Although, the conditional loss of RBPJ produced a similar reduction of EVP and self-renewal capacity, there was a significantly different and often opposing fate resulting from EVP differentiation towards mesenchymal cells. This was also reproduced upon conditional $Ptch1$ deletion. Overall, these findings highlight the importance of Sox9, Notch and HH pathways in EVP function. However, loss of these pathways results in unique changes in cell fate.

The opposing phenotypes observed upon conditional loss of $Sox9$ or $Rbpj$ supported a negative regulatory role between these two pathways. Indeed, in wounds, $Sox9$ deletion resulted in decreased M cells emanating from EndMT whereas Rbpj deletion increased mesenchymal transition and scarring. At the molecular level both in vitro and in vivo, negative regulation of canonical

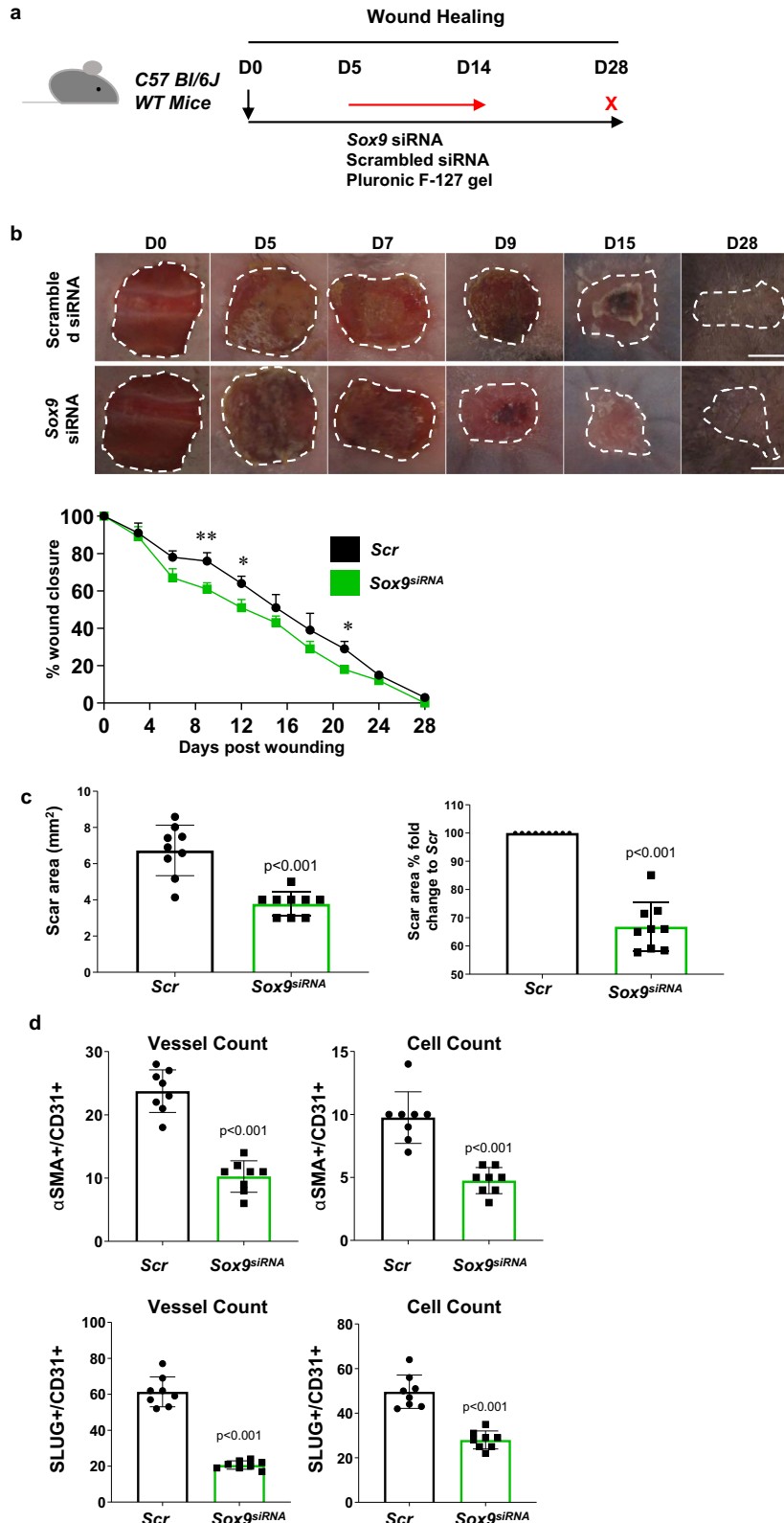

**Fig. 8 Reduced scarring upon use of siRNA to inhibit *Sox9* in wound healing. a** Schematic diagram demonstrating experimental set up using wild-type mice. **b** Photo images showing wound healing over the course of 28 days (D28) post wounding between scrambled (*Scr*) controls and *Sox9* siRNA (*Sox9^siRNA^*) treated mice. Significant increase in wound healing was observed in mice treated with (*Sox9^siRNA^*) compared to *Scr* (*$p < 0.05$; **$p < 0.01$ vs *Scr*; $n = 8$ biologically independent animals; mean ± SD; $p$ value was calculated by two-way ANOVA with multiple comparison of row mean). **c** Significant decrease in macroscopic scar area was observed in mice treated with (*Sox9^siRNA^*) compared to *Scr* (**$p < 0.01$ vs *Scr*; $n = 8$; mean ± SD). **d** Quantification of skin wound sections at day 7 (D7) with endothelial markers CD31 and EndMT markers SLUG and αSMA (**$p < 0.01$; ***$p < 0.001$ vs *Scr*; $n = 5$ biologically independent animals; mean ± SD; $p$ value was calculated by two-tailed unpaired $t$ test).

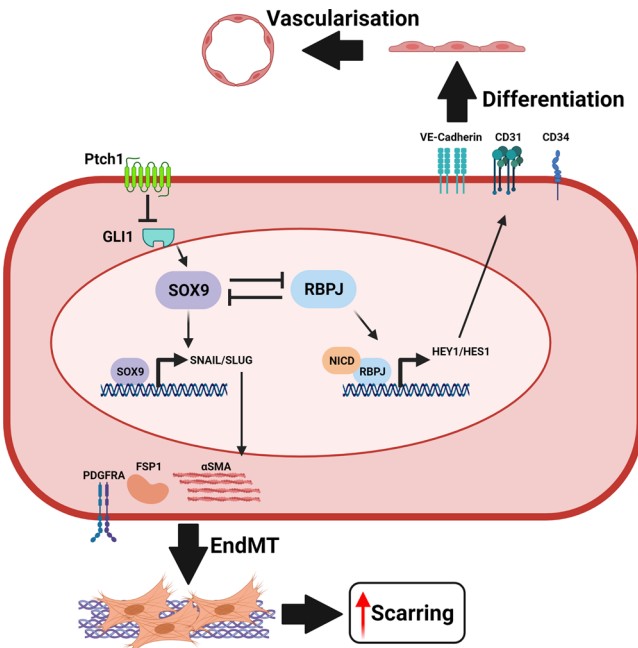

**Fig. 9 Regulation of endovascular progenitors cell fate.** The fate of EVPs is maintained by the co-inhibitory relationship between SOX9 and RBPJ maintained in a progenitor state in homoeostasis. However, increased expression of SOX9, either through the loss of RBPJ, or activation of the hedgehog pathway results in the translation of SNAIL and SLUG, driving the expression of mesenchymal-related factors such as FSP1, αSMA and PDGFRA. Thus, triggering EndMT and excessive scar formation. Conversely, the deletion of SOX9 promotes Notch signalling and RBPJ activities, favouring endothelial differentiation and vascularisation.

Notch signalling increased Sox9 expression at RNA and protein level. Similarly, *Sox9* deletion in vivo resulted in increases in canonical Notch signalling. However, further increase in this signalling did not result in Sox9 reduction, possibly suggesting that Sox9 gene negative regulation is exerted by RBPJ alone. This duality could be further visualised in the *Ptch1* conditional ablation model. Specifically, the mesenchymal population upregulated Sox9 and fibrosis-associated TFs and downregulated Notch signalling as well as genes associated with endothelial fate. In contrast, in the same mice, ECs had high levels of Notch signalling and had downregulated *Sox9*. It remains unknown whether the HH effectors, *Gli1* and *Gli2*, directly or indirectly regulated target genes.

The opposing fates adopted by ECs in the context of wound healing could be defined by markers of Notch signalling, *Hey1* and *Hes1*, which have been extensively described in human and murine endothelial biology. Specifically, Notch signalling integrates with cyclin-dependent kinase inhibitors p16, p21 and p57 for quiescent progenitor maintenance, and endothelial attachment genes including CD31 and VE-Cadherin to maintain endothelial function[22,25]. Notably, the downregulation of Notch signalling in this study revealed a loss of endothelial self-renewal characteristics and endothelial quiescence genes such as *Il33*[25]. In contrast, upregulation of EndMT-associated genes in the transitioning mesenchymal population further elucidates the contribution of HH-activation to EndMT by indicating transcriptional biomarkers for a more complex mesenchymal transition. For example, *Snail* has predominantly been associated with pathological fibrosis downstream of the classical TGFβ pathway, whereby the recruitment of *Twist* transcriptional regulators drive the further upregulation of motility genes such as *αSMA* and matrix metalloproteases, and excess deposition of fibrotic collagen

subtypes[34,38–40]. Furthermore, Snail stabilised by GSK-3β is demonstrated to induce osteogenic EndMT contributing to atherosclerotic calcification in response to the cardiovascular disease risk factor, oxLDL[3,41]. Our group has previously identified the notion of a resident endothelial progenitor cell having probable biopotential capacity[9]. Shafiee et al., identified a novel meso-endothelial cell from term placental tissue that demonstrated Notch signalling to be crucial in maintaining endothelial phenotype. In contrast however, differentiation to a mesenchymal phenotype was not via the classical TGFβ EndMT pathway. This has also been previously shown in situations of EndMT, whereby Slug is activated independently from TGFβ and Snail directly by Notch signalling pathway in the initiation of cardiac cushion EndMT[42–44]. Slug function is further dependent on cooperative Sox9 protein-binding to induce TGFβ-mediated EndMT progression, consistent with the upregulation of Sox9 in EndMT as identified here[45]. Previous literature has reported an interdependent protein interaction between SLUG and SOX9 in mammary stem cells, human lung carcinoma and kidney fibrosis[27,45,46], whereby inhibition of either SLUG or SOX9 resulted in inhibition of the mammary bipotent stem cells or cancer cells[27,45]. This highlights the complexity of bipotential differentiation, therefore elucidating differentiation pathways will aid in better understanding tissue regeneration and/or disease initiation.

Currently, there are over 20 clinical trials utilising siRNA to silence aberrant genes in a variety of pathologies including cancer, atherosclerosis and hypertrophic scarring[47]. Similarly, *Sox9* siRNA has been previously shown to be efficient in the knock-down of *Sox9* in cancer and other pathologies[48,49]. To administer the siRNA, Pluronic F-127 gel was utilised, which has proven effective in wound healing and drug delivery[50,51]. A significant reduction was detected in fibrotic scar area following *Sox9* siRNA application in comparison to a scrambled siRNA administered control. *Sox9* plays a vital role in the skin wound healing process due to its regulation of HFSCs (hair follicle stem cells), re-epithelisation of the wound tissue and stimulating ECM secretion via regulation of fibroblasts and myofibroblast populations[34,47,52,53]. In this study, we explored the possibility of a preventive therapy to avoid excessive scarring. It is unclear if *Sox9* siRNA and inhibition of EndMT will have any impact on already established scars.

In conclusion, *Sox9* expression in EVP is essential in maintaining quiescence and clonogenicity, however is also essential in driving pathological EndMT when a loss of *Rbpj* and Notch signalling is observed. Our findings strongly suggest that blocking *Sox9* within EVPs is a valid strategy in reducing EndMT and vascular fibrotic disease.

## Methods

**Animals**. All mice were maintained in accordance with University of Queensland ethics approvals and guidelines for care of experimental animals. Mice, regardless of sex (ages 10–12 weeks of age; genders housed separately) were used for this study. *C57BL/6* mice (WT) were obtained from the Animal Resources Centre (Perth, Western Australia). All mice were group-housed (maximum of five mice per cage) and maintained under a regular light–dark cycle altered every 12 h with free access to water and standard mouse chow. The *Cdh5-CreERt2* mouse line was crossed with ROSA$_{lox}$YFP$_{lox}$ for endothelial-specific lineage tracing experiments. The subsequent double transgenic offsprings were termed *Cdh5-CreER RosaYFP*. To induce endothelial specific knockout of the gene *Sox9*, *Rbpj* and *Ptch1*, *Sox9*$^{fl-fl}$, *Rbpj*$^{fl-fl}$ and *Ptch*$^{fl-fl}$ mice then bred with *Cdh5-CreER RosaYFP* to create the triple-transgenic offsprings *Sox9*$^{fl/fl}$/*Cdh5-CreER RosaYFP*, *Rbpj*$^{fl/fl}$/*Cdh5-CreER RosaYFP* and *Ptch*$^{fl/fl}$/*Cdh5-CreER RosaYFP*, respectively. In all lineage tracing experiments, mice were subjected to a regimen of tamoxifen injection up to 10 days. Tamoxifen (Sigma-Aldrich, MI, USA) was dissolved in 90% corn oil (Sigma-Aldrich) and 10% ethanol, with each mouse receiving a 2 mg dose daily in 100 μL.

**In vivo wound healing assay**. Full thickness excisional wounds were created using a 6 mm sterile punch biopsy (Stiefel Laboratories, Research Triangle Park, NC) along the dorsal skin. Wounds were then left open with animals culled at each

respective timepoints after wounding. In lineage tracing experiments, animals were injected with tamoxifen prior to wounding. Wound closure and total surface were manually defined, with the surface area recorded at individual timepoints. Measurements for the surface area were calculated using ImageJ software (National Institute of Health). To assess fibrosis and scar formation, large full thickness excisional wounds ($1.5 \times 1.5$ cm) were created. The wounds were left to heal for 28 days before the scar tissues were harvested and measured for scar area using ImageJ.

**Skin wound harvest and processing**. Mice were culled using $CO_2$ asphyxiation. Organs and 6 mm wounds with 2 mm of surrounding skin were dissected and fixed in 4% paraformaldehyde (Sigma-Alrich, MO, USA) for 2 h at 4 °C. Fixed samples were cyro-protected with incubations in 10% and subsequently 30% sucrose (Chem-Supply, SA, AUS) in phosphate buffered solution (PBS) overnight. The samples were embedded in Tissue-Tek Optimal Cutting Temperature (Sakura Finetek, CA, USA) in preparation for cyro-sectioning. Before embedding, wounds were cut in half to ensure that sections were taken from the centre of the granulation tissue, whereby 10 μM sections were taken at 30 μM intervals onto Superfrost Plus slides (Thermo Fisher Scientific, Waltham, USA) using a CM1905 Cyrostat (Leica Biosystems, Wetler, Germany). Slides were stored at −30 °C.

**Measurement of large wound scar surface area**. Tile-scan images of scars were conducted using an Olympus BX51 Fluorescence microscope (Olympus, Tokyo, Japan) (bright-field) at 4X magnification, with the edge of the hairline taken as the edge of the scar. The scar surface area was measured in ImageJ and a fold change was calculated using the control.

**PicoSirius Red staining and analysis**. Wounds were stained using PicoSirius red staining (Invitrogen, CA, USA) as previously published[9]. Tile-scans of scars were imaged using an Olympus Upright Epi (Olympus, Tokyo, Japan) (bright-field) at 10X magnification. Images were analysed in Visiopharm (Visiopharm, Hørsholm, Denmark) software, where scar area was calculated using the in-built 'tune' tool, allowing the software to be taught to recognise the scar area, total scar area was then calculated.

**Immunofluorescence**. Tissue collected were fixed for 2 h at room temperature in 4% PFA. This was followed with 3x wash of 1x PBS (Amresco, Solon, Ohio, USA). Tissues were then infused under a sucrose gradient for cryo-protection before cryo-embedding in Tissue-Tek® O.C.T. Compound (Sakura Finetek, Torrance, California, USA). Prior to antigen staining, cryo-sections were permeabilized in 0.5% TritonX-100 (Chem Supply, Gillman, South Australia) before blocking with 20% normal goat serum. In this study, primary antibodies used includes rat anti-mouse CD31 (1:100), rabbit anti-αSMA (1:200), rabbit anti-FSP1 (1:200), rabbit anti-SLUG (1:200), rabbit anti-SOX9 (1:500), rabbit anti-ERG (1:500) and mouse anti-SOX9 (1:200). Primary antigen staining was conducted overnight at 4 °C. Sections were then subjected to $3 \times 5$ min washes in 1x PBS + 0.1% Tween-20 (Amresco, Solon, Ohio, USA). Secondary antibodies conjugated with Alexa-fluor 568 or 647 (Invitrogen, Carlsbad, CA, USA) were used for fluorescence detection. Sections were incubated with secondary antibodies for 40 min at room temperature. Nuclear staining was revealed in specimens mounted with ProLong® Gold mounting media containing DAPI (Invitrogen, Carlsbad, CA, USA).

**En face aorta preparation**. For en face aorta staining, mice were perfused with ice-cold 1x PBS, then 4% PFA and a final perfusion of PBS. After, aortas were removed for further dissection and cleaning. Under a dissection microscope, attached diaphragm, vascular branches and perivascular adipose tissue were removed from the aorta. The aorta was then opened via longitudinal dissection by spring scissors and flatten with minutien pins onto dissection wax (Fine Science Tools, United States). Flatten aortas were fixed for 2 h at room temperature in 4% PFA. The fixative was removed with 3x washes of 1x PB, followed by blocking with 20% normal goat serum in 1x PBS. For en face staining, primary antibodies used included rat anti-mouse CD31 (1:100), rabbit anti-SOX9 (1:200) and rabbit anti-activated NICD (1:200). Subsequent secondary staining and mounting was carried out as described in the previous immunofluorescence section.

**Microscopy**. Confocal images were acquired with either the Olympus FV3000 confocal microscope or CSU-W1 SoRa Nikon Spinning Disk microscopes using Flouview FV3000 software v1.0 and NIS imaging software v4.5, respectively. Confocal images were acquired with a microscope equipped with Argon 561–10 nm DPSS and 633 nm HeNe lasers, and a 405–30 nm diode. Images were obtained at 20x, 40x and 100x. Immunofluorescence vessel quantification was conducted using ImageJ v1.53 and NIS elements v4.60.00. PSR collagen quantification was conducted using Olyvia v3.2.1.

**Administration of siRNA**. Sox9 and scrambled siRNA (control) (Dharmacon, VIC, AUS) were diluted to 3 μM in Ultrapure distilled water (Table S1 for siRNA sequence). A 30% w/v solution of Pluronic® F-127 (Sigma-Aldrich, MO, USA) and siRNA was made and administered twice daily directly onto large and small

wounds using a pipette. In total, 75 μl was administered to each large wound and all four small wounds. For large wounding experiments, mice started the siRNA administration from D5 to D12. For small wounding experiments, mice started the siRNA administration from D1 to D7.

**Tissue processing and digestion**. Tissues were collected for FACS analysis at defined specific end timepoints (D5 and 7). Aorta and skin wounds were first finely minced with scissors and digested for 20 min at 37 °C in an enzymatic cocktail containing 1 mg/ml collagenase I (Gibco, Life Technologies, NY, USA), 1 mg/ml dispase (Gibco, Life Technologies, NY, USA), 150 μg/ml DNase-I (Sigma-Aldrich, St Louis, MO, USA) in Hanks' Balanced Salt Solution. After digestion, the cell suspension was then passed through a 70 μM cell strainer. Single cell suspensions were then used for FACS sorting or analysis by flow cytometry.

**Flow Cytometry and FACS**. Dissociated single cells in FACS buffer (0.5% BSA, 1 mM EDTA in 1x PBS) were then incubated with various antibody combinations for FACS analysis and sorting. A Gallios™ flow cytometer was used for sample acquisition, with subsequent data analyses performed with Kaluza® analysis software (Beckman-Coulter, Miami, Florida, USA). A FACSaria cell sorter was utilised for FACS sorting using the FAVSDiva v5.0.3 software with subsequent analysis performed with FlowJo v10 (Becton Dickinson, Franklin Lakes, NJ, USA). Standard doublets discrimination and 7′AAD or FVS live/dead staining was carried out to only include live singlet events. The following combinations of antibodies were used to assess the endothelial hierarchy populations: Rat anti-mouse VEGFR2 PE (1:100), CD31 PE-Cy7 (1:1000) and CD34 Alexa647 (1:100) (Becton Dickinson, NJ, USA), Rat anti-mouse Lineage cocktail BV450 (1:50) (Biolgend), CD26 PE (1:200), CD140α BV605 (1:100) and Rat anti-mouse CD144 FITC (1:50) (eBioscience).

**Cytospin of FACS sorted cells**. Cells are FACS sorted directly into 2% FCS, and then washed with and resuspended in 1X PBS.A Cytospin™ 4 Cytocentrifuge was utilised for the distribution of cells in suspension onto SuperFrost Plus™ slides. In short, cells were loaded into supplied columns and inserted into the centrifuge as per previously published protocol[54]. Adhered cells were then fixed with ice-cold 4% PFA and then blocked with 20% NGS. Standard immunofluorescent staining protocol described in the previous section was then carried out.

**RNA Extraction, cDNA synthesis and qPCR**. RNA was extracted from FACS sorted or cultured cells using a QIAGEN mini kit (Qiagen, Valencia, CA) according to the manufacturer's instructions. RNA quality and concentration was assessed using A260nm/A280nm spectroscopy on the Nanodrop ND-1000 (Thermo-scientific, Langenselbold, Germany). About 5-100 ng of RNA was used for cDNA synthesis using the Superscript III Reverse Transcription Kit (Invitrogen, Mount Waverley, Australia). For cDNA synthesis, collected RNA was treated with DNase-I (Sigma-Aldrich) to remove genomics DNA contamination. After which, cDNA synthesis was conducted using the Invitrogen SuperScript III First Strand Kit (Life Technologies, United States). cDNA was then treated with RNase H (Life Technologies). To quantify relative changes in key gene expression after treatment, real time-qPCR was conducted on cDNA using SYBR Green Master Mix reagent (Applied Biosystems, United Kingdom). Fold change of gene expression was determined using the delta delta Ct method, normalised to housekeeper gene Rpl5 in mouse or HPRT1 in human. Primer sequences are listed in Supplementary Table 2.

**ECFC culture**. Human term placentas were obtained after informed written consent from healthy pregnant women and placental foetal ECFC and MSC isolated using our previously published protocol[24]. The use of human tissue was granted by the human ethics boards of The University of Queensland and the Royal Brisbane and Women's Hospital. ECFCs were cultured on rat tail collagen coated tissue culture flasks in Endothelial Growth Medium (EGM-2, Lonza Group, Basel, Switzerland) with 10% of foetal bovine serum (FBS). MSCs were cultured using standard tissue culture flasks in DMEM with 10% (FBS).

**Western blots**. Protein were obtained from cells lysed with radio-immunoprecipitation (RIPA) buffer supplemented with 1x complete protease inhibitor (Sigma-Aldrich) followed by sonication and centrifugation. SDS-PAGE of lysates was performed with the Bolt™ 4 to 12%, Bis-Tris system (Invitrogen) and protein transfer was conducted with the Biorad Trans-Blot Turbo Transfer System according to the manufacturer's protocols. Immunoblots were labelled with following primary antibodies after blocking with Licor Intercept blocking buffer: rabbit anti-SOX9 (1:1000, Merck Millipore, Burlington, Massachusetts, USA), rabbit anti-activated NOTCH1 (1:1000, Abcam) and mouse anti-β-actin (1:5000, Sigma-Aldrich). Primary antibodies were detected using fluorescently conjugated secondary antibodies: goat anti-rabbit IgG IRDye® 800CW and goat anti-mouse IgG IRDye® 680RN (1:2500, LI-COR Biosciences, Lincoln, Nebraska, USA). Detection and quantification of fluorescence intensity were performed using an Odyssey CLx imaging system (LI-COR Biosciences, Lincoln) and Odyssey 2.1 software or ChemiDoc MP Imaging System and ImageLab software v6.1. For an example of presentation of full scan blots, see Supplementary information.

**Statistical analysis and reproducibility**. All statistical analyses were performed using GraphPad Prism v8 software. Data were analyzed using the following tests: two-tailed unpaired *t*-tests one-way or two-way ANOVA with Bonferroni correction. A *p* value <0.05 was considered significant. For immunofluorescent and macroscopic quantification of granulation tissue, at least five slides were evaluated per wound, collected from a minimum of five biologically independent animals. Quantification of protein expression in the aortic endothelium was carried out with a minimum of five aortas from biologically independent animals. Quantification of protein expression from cytospun cells were carried out on cells sorted from a minimum of three groups of three biologically independent animals. Data were averaged per biological replicate for statistical analysis. For all quantification, single blinding was conducted.

**Reporting summary**. Further information on research design is available in the Nature Research Reporting Summary linked to this article.

## Data availability
The data that support the findings of this study are available from the corresponding author upon reasonable request.

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

## Acknowledgements

We would like to thank the flow cytometry facility of the Translational Research Institute for their assistance in the many cell sorting experiments undertaking. The study was funded by the Australian Research Council (ARC) Discovery Project (DP190103187). J.P. salary was supported by ARC Discovery Early Career Research Award (DE180100984). K.K. salary was supported by NHMRC Career Development Fellowship (APP1125290).

## Author contributions

J.Z., J.P., M.F., M.C.Y., K.K.: acquisition of data, analysis and interpretation of data, obtained funding for the study and drafting of the manuscript. J.P., J.Z., I.H., S.K., H.H., J.D., H.Y.W., S-.L.S. and G.H.: acquisition of data and analysis. All authors have read the manuscript and approved it for publication.

## Competing interests

J.P. and K.K. are co-inventors on a patent relating to the isolation of endothelial progenitors from the human placenta. The remaining authors declare no competing interests.
