## [Peer Review File · Nature Communications]

Reviewers' Comments:

Reviewer #1:

Remarks to the Author:

In the manuscript by Patel et al., the authors continue their quest to better understand a population of endothelial progenitors (EVPs), which they have defined in previous work. This report is focused on the role of two transcriptional regulators (SOX9 and RBPJ), and their role in EVP dynamics and wound healing response, using genetic mouse models. They also venture into the role of Hedgehog signaling, using mouse models where Hh signaling was elevated by ablating Patched in the endothelium.

The major conclusion is that SOX9 and RBPJ act as negative regulators of each other. Endothelial depletion of SOX (through a Chd5-CRE system), led to EVP depletion, elevated of RBPJ and Notch signaling as well as reduced skin scarring (reduced EndMT).

Specific comments:

Line 126-127: The authors find that SOX9 immunostaining is only observed in EVP and not in D cells. This finding is interesting, and should be addressed further (or at the very least discussed) in the light of that SOX9 mRNA is expressed widely across both arteriolar, venous endothelial cells as well as in capillaries (see recent scRNA-seq atlas projects). Although one cannot exclude selective translation, this expression pattern is difficult to reconcile with SOX9 being expressed only in EVPs and not more widely in the endothelium. If SOX9 distribution would be more widespread in the endothelium, the conclusion from the lineage tracing experiments in wound healing would be in doubt. The true endothelial distribution could be more strictly tested in a Sox9-CRE lineage tracing model.

I accept the authors' view that loss of SOX9 elevates RBPJ expression, and conversely, loss of RBPJ leads to enhanced SOX9 expression, which is in its own right an interesting finding. The postulated reciprocal negative regulation between SOX9 and RBPJ would however implicate opposite phenotypes in the EPV/D cell transition and wound healing settings. Notably, however, in both the SOX9 and RBPJ KO models, there is a depletion in EPVs, and the authors do not provide a satisfactory explanation for this, which is a bit disappointing, as this is critical for their "story". Furthermore, the response of endothelial quiescence genes (IL33, p16, p21, p57) was similar in both KO models.

As a follow up on the comment above about, it is a bit surprising to see that there are no data on the wound healing effects of RBPJ, in particular as the authors have already generated the necessary mouse crosses to conduct the analysis. It would have been interesting to learn about the effect of RBPJ ablation in the vasculature using their skin wound model in the light of their hypothesis above.

Similarly (and which would not even require the transgenic mice), it would have been nice to see siRNA experiments for RBPJ in parallel with the SOX9 siRNA experiments, which appear to give a rather distinct phenotype. Without these experiments, I am worried that the SOX9-RBPJ hypothesis is not fully validated.

There are also a number of shortcomings on the Notch side:

The authors equate Hes and Hey regulation with regulation of the Notch response. Additional markers (Nrarp, cMyc) etc could have been used.

The authors report upregulation of RBPJ as well as Hes/Hey in the SOX9-deficient cells. It is not a given that Hes and Hey will increase because of more RBPJ in the cell. In the Notch field, there are ample examples that elevated RBPJ can lead to repressed Notch activation, as RBPJ is a repressor when Notch is not activated. This would need to be discussed in more detail.

Minor comments:

In a number of places, there are comments referring to previous literature but with no matching

citation, e.g. "SOX9 in this pathway classically attributed to TFBF signaling" (line 227). Improved referencing would be warranted.

The Discussion is generally a bit "all over the place", discussing in general terms a broad set of subjects rather than focusing on discussing at depth the data in the report. As an example, the SOX9-Notch link could have been discussed in the context of previous reports where SOX9 has been shown to be a downstream genes in the Notch pathway (see e.g. data from Sean Morrison's lab (this reviewer is also not Morrison). A better review and discussion of the existing literature would make the Discussion more interesting.

The authors frequently use terms like "strikingly", "importantly", "we clearly observed". These exclamatory remarks gives a "tabloid" ring to the text, which is unnecessary; the data are clear anyway.

Reviewer #2:

Remarks to the Author:

In this manuscript, the authors utilize knockout mouse models to elucidate the role of the transcription factor Sox9 in regulating EndMT (Endothelial to Mesenchymal Transition) during skin wound healing. Based on RNA-Seq data published in a previous study (Prudence D et al., 2019), the authors find and confirm that Sox9 is highly expressed in EVP (Endovascular progenitors) in homeostasis and wound healing. Knocking out Sox9 specifically in the endothelium, using the Cdh5-CreER driver, led to a reduction in the number of EVP and an increase in D cell number. This was accompanied with both impaired EndMT and vascularization, resulting in accelerated wound healing and less scarring. To delineate the mechanism of Sox9 regulation of EndMT and its connection with different signaling pathways, the authors use additional knock-out mouse models to first, target Rbpj and Notch signaling and second, activate Hh signaling in the Cdh5+ endothelial cell population. In this manuscript, the authors provide several interesting insights for the mechanisms that regulate EndMT and their role in wound healing and scar formation.

Major comments:

1. The authors argue that Sox9 regulates EVP maintenance and represses their differentiation to D cells. Deletion of Sox9 in the endothelium reduces the likelihood of both endothelial vascularization and EndMT process, resulting in decreased scarring. From qRT-PCR results of the sorted EVP, the authors find that several genes related to Notch signaling are significantly upregulated. The RbpjeKO shows a similar phenotype of decreased EVP cell number as well as increased D cell number. The deletion of Rbpj in endothelial cells promotes EndMT and scarring but does not induce vascularization. If Sox9 and Rbpj interact via a negative feedback loop, how do the authors explain that deletion of Rbpj gene does not cause more vascularization?

2. When the authors analyze gene expression differences in the Sox9 KO mice, they use sorted EVP cells in both control and Sox9 KO, but when comparing the differences in RbpjeKO mice they use the whole endothelial population. The observed upregulation of Notch signaling in EVP cells from Sox9eKO mice is thus not directly comparable to the upregulation of Sox9 in whole endothelial cells. Therefore, this weakens the argument that Sox9 and Rbpj interact via a negative feedback loop in regulating EndMT process. Additionally, both knockout mice show the similar trend in decreased number of EVP cells.

3. The authors propose that Hh signaling plays a critical role in the EndMT process.

The authors start to compare gene expression in CD34+ and CD34- populations in both control and Ptch1eKO mice. They show that Sox9 is expressed in EVP (CD34+) cells but surprisingly they also show that Sox9 is expressed in CD34- cells as well. This is important and was not mentioned

previously in the manuscript. The authors did not perform a similar analysis of Sox9 expression in the Sox9eKO and RbpjeKO mouse models, to test whether there are differences in gene expression between CD34⁻ and CD34⁺ cells.

Furthermore, their results show that in the Patch1eKO mice the expression pattern of the endothelium (CD34⁺) resembles that of the Sox9eKO but also that CD34⁻ cells have similar expression (Sox9 up, Notch down) to the endothelium in RbpjeKO. The scarring phenotype is similar between Patch1eKO and RbpjeKO. Does this mean that the CD34⁻ population takes over the role of CD34⁺ population-EVP cells and becomes the key population for EndMT to promote scarring? The authors provide complicated data but they do not identify clearly what is CD34⁻ population and whether Sox9's function in EndMT process is intergrated with or regulated by Notch signaling or Hh signaling.

4. Previous studies have shown that Sox9 gene is critical for hair follicle stem cell maintenance. Conditional deletion of Sox9 in hair follicles triggers the differentiation of hair follicle keratinocytes to epidermal cells (Meelis Kadaja et al., 2014), which may facilitate the wound healing. Here the authors perform Sox9 siRNA topical treatment after wounding which may also affect Sox9 expression in hair follicle stem cells. Given the broader implications of this treatment the authors should analyze the histology of hair follicles and the kinetics of the hair cycle. Also, keratinocyte proliferation and apoptosis should be assessed after topical treatment, to make sure the phenotype is not caused by the changes of keratinocyte activity rather than the endothelium. The author mention this in the discussion but did not provide any data.

Other issues:

1. The authors should test more mesenchymal markers, such as α -SMA, FSP-1, vimentin and Col1a2, not only by qRT-PCR, but also by IF. The authors should also provide more Immunostaining data of EndMT markers, snail and α -SMA, co-staining with CD31.

2. The resolution of Sirius red staining data is low; it is hard to evaluate the difference between wildtype and Sox9eKO mice.

3. The authors create 1.5 cm X 1.5 cm wounds to assess fibrosis, but all the flow data are made from 6 mm punch wounds. It is an open question whether the pattern of EVP and D cell changes are consistent in these different wound scenarios.

4. In figure 3C, is wound healing also affected from day 4 to day 12?

6. The authors broadly define the CD34⁻ population as mesenchymal cells. I feel that additional surface markers, like α -SMA, S100a4, are needed to better characterize this population.

7. For figures 4C, 6C, 6D and 7D, the authors should also provide representative high resolution immunostaining data.

8. To confirm Sox9 upregulation in the different cell populations, the authors should demonstrate Sox9 expression by immunostaining in the RbpjeKO and Ptch1eKO mice after wounding, not just by RT-PCR.

9. The phenotype of RbpjeKO mice was published previously by the same author, so this part is less novel.

Reviewer #3:

Remarks to the Author:

Patel and colleagues set out to examine the role of Sox9 and Rbpj signaling on endothelial progenitor cells during tissue repair. The rationale for these studies is that endothelial-to-mesenchymal transition is a major contributor to fibrosis during skin wound healing. Although this may be true of cardiac wound healing, I am not convinced that this has been definitively shown for skin and this manuscript in its current form does not directly support that claim any further. The authors employ mouse models in which they genetically delete either Sox9, Ptch or Rbpj within "endothelial progenitors" and subsequently evaluate the impact of each on quality of skin wound healing. The authors provide evidence suggesting that intriguingly, loss of Sox9 in endothelial progenitors results in reduced scar formation and this is also corroborated by topical application of siRNA against Sox9. Although interesting, the authors fail to provide definitive evidence from their fate mapping experiments. Although they provide cytometry data, there is a lack of imaging showing specificity of labeling within cutaneous endothelial progenitor cells or their fates following skin injury. It would also be important to show that these same genes (Sox9, Ptch, Rbpj) are co-expressed in the putative endothelial progenitors in vivo during homeostasis and following injury. Although I think the manuscript is quite interesting and would be of importance to the wound healing community, there are a number of shortcomings that need to be addressed (described below).

1. Figure 1 – The authors need to show confocal images of skin sections from Cdh5Cre mice that demonstrate a) exclusive expression in endothelial cells, but not other dermal cell types, b) co-expression with Sox9 and c) verification with additional endothelial markers. I do not understand the rationale for doing in vitro immunostaining of cells that have been isolated from acute skin wounds. If the authors are going to claim that "Sox9 expressing endothelial progenitors" are contributing to skin wound healing then they need to demonstrate 1) their presence in normal skin vasculature and 2) then show active recruitment into the wound (and differentiation to myofibroblast phenotype within the wound). Providing a kinetic analysis at different times post-injury would be an essential step in defining their contribution to wound healing and active contribution to scar formation/neodermis.

2. "Additionally, skin wound analysis from Sox9 conditional knock-out mice demonstrated a significant reduction in pathological EndMT resulting in reduced scar area."

a. What was the impact on myofibroblast numbers in the wound? This needs to be quantified.
b. How do you know that you haven't just accelerated/increased contraction of the wound rather than actually mitigating fibrosis? Can you measure overall contraction of the wound in each genotype? This is particularly important given that the mouse skin healing is primarily driven by contraction and you have not splinted the wounds to specifically look at secondary intention. Histological images of the wounds would be helpful to understand the overall impact on wound outcomes. Inclusion of second harmonic imaging to characterize the collagen content/orientation would also be useful. Cd26 staining alone is not indicative of a 'fibrotic' fibroblast.

3. It is not clear to me why you are examining "human placental fetal ECFC and MSCs" – this seems entirely extraneous. Given the focus is on skin wound healing, the authors should focus their analysis to the vascular endothelial cells within the skin.

4. Similarly the analysis of endothelial cells from aorta in this manuscript is again of little relevance. All of the analyses currently performed using aorta should be done on cutaneous endothelial cells.

5. When do the EPV (or their derivatives) appear within the skin wound? Where do they go? How many of them relative to mobilized dermal fibroblasts? Showing kinetics and images of cells within at different times following injury is essential to underscore the importance/contribution of these cells to the wound healing process.

6. As it stands, the title of the paper is not well supported by the data presented.

7. Why have you chosen to use two different skin wound models (6mm diameter excision versus

1.5 x 1.5cm)? Using a wound size of 1.5 x 1.5 cm, it would be expected that there would be some level of hair follicle neogenesis. Was this observed? This might be a better proxy for reduced fibrosis and increased regeneration in any of your genetic KO models.

8. Is there a difference between the EPV contribution to your 6mm wound and 1.5cm wound ?

9. The measures used for fibrosis are quite limited (e.g. picosirius red quantification and Cd26 – presence of cd26 in skin is not definitive of a fibrotic phenotype given it is expressed in uninjured papillary dermis and hair follicle mesenchyme). This should be expanded upon. How many sections per wound are you measuring across the wound and from where are you sampling? This needs to be made clear. Inclusion of second harmonic generation to measure collagen content and orientation, dermal thickness, dermal cell density would be much more convincing.

Text comments:

Lines 123-128 : Please show the FACS strategy in the figures.

Lines 129-130 : “We have reported the accumulation of EVP” – Can you please describe the relevance of this statement?

Line 132 : Cdh5 : show in skin to trust the specificity of Cadherin 5 in skin or ref ?

Line 135 : What is the center of the wound at D1 ? How is it possible to have cells in the center of the wound, except immune cells and blood ? Images of the skin wounds at Day 1 need to be shown to verify labeled cells in the centre of the wound.

Lines 138-141 : add a schematic to explain the different populations

Lines 157-158 : It would be helpful to add a schematic to explain EPV, D, TA and FACS strategy

Lines 166-175 : How did you assess that ? Brdu ? EdU ? When did you inject the DNA labeling ? This does not appear to be described in the material & methods.

Line 186 : Picosirius red staining: the staining doesn't seem appropriate, as if the tissue sections are folded. Shouldn't you see 3 distinct skin layers within the normal skin? Higher magnification images should be included to verify the rigor of the quantification.

Lines 199-201 : Explanation for inclusion of SLUG staining should be made clear. Why are you not showing aSMA images along with the SLUG ?

Lines 204-207: show higher magnification images. What is the significance of CD34+/-, CD31+/-?

Lines 357-358 / 362-364 : “blood vessel formation or to enter Mesenchymal transition”.

Unfortunately, you never show that. If you want to keep this title and this interesting conclusion, you should show a skin wound with YFP+ cells into the dermis, not in the vessels and collagen productive cells, for example

Lines 784 – 785: should be μ and not μ M

Lines 867-879: you should explain which method you use for qPCR quantification

General figure comments:

1. Your Y labels are not clear at all in all the figures (percentage of what ? fold change of what % of gated ?)

2. For all of your qPCR controls and the percentage of scar area you don't include an indication of variance (SEM or SD)? Please show this!

3. Please add n-values for # of mice used in each experiment and indicate this for each analysis in the figure legends.

Specific comments on figures :

- Figure 1: Not clear if we are in skin / aorta ? Please add a schematic

- Change the order in the figures 4D and 4E-F

- Figure 3-C : the wound closure in LOW is done around at D14-D16. How do you evaluate this percentage?

- Fig 3D,E – The imaging provided is entirely unconvincing. How can you actually measure anything from these images? Additional images and details of how this was exactly quantified will need to be provided.

- Figure 3E: high magnification or better image?

- Figure 4-C: Why do these different vessel signatures are different?

- Fig S6 : should be in the main figures

- Fig S1 : Figure S1B "We also observed that the SOX9 staining is localised only to the endothelial layer and not in the underlying vascular smooth muscle layers " – What is the red staining? Why are you not showing images of Sox9 labeling in the deeper image slices? This set of panels is confusing and needs to be redone.

- S2A: This is surprising that you don't see anything in the KO – can you explain this?

- S2B. Why are you not showing YFP fluorescence?

-

REVIEWER COMMENTS

Reviewer #1 (Remarks to the Author):

In the manuscript by Patel et al., the authors continue their quest to better understand a population of endothelial progenitors (EVPs), which they have defined in previous work. This report is focused on the role of two transcriptional regulators (SOX9 and RBPJ), and their role in EVP dynamics and wound healing response, using genetic mouse models. They also venture into the role of Hedgehog signaling, using mouse models where Hh signaling was elevated by ablating Patched in the endothelium.

The major conclusion is that SOX9 and RBPJ act as negative regulators of each other. Endothelial depletion of SOX (through a Chd5-CRE system), led to EVP depletion, elevated of RBPJ and Notch signaling as well as reduced skin scarring (reduced EndMT).

Specific comments:

Line 126-127: The authors find that SOX9 immunostaining is only observed in EVP and not in D cells. This finding is interesting, and should be addressed further (or at the very least discussed) in the light of that SOX9 mRNA is expressed widely across both arteriolar, venous endothelial cells as well as in capillaries (see recent scRNA-seq atlas projects).

We thank reviewer for highlighting the new findings in this area. Although Sox9 is differentially expressed between EVP and D cells we find its gene expression at mRNA level to be detectable in D cells as well (page 6; line 110-114). In this instance our findings do not contradict the widespread expression of Sox9 at mRNA level. We have now added a reference to existing atlas work supporting our findings of Sox9 expression in the endothelium (page 6; Line 115)

Although one cannot exclude selective translation, this expression pattern is difficult to reconcile with SOX9 being expressed only in EVPs and not more widely in the endothelium.

We have now conducted more systematic and quantitative analysis of Sox9 expression in flow sorted EVP versus D cells (page 5; line 128-132). This quantitative analysis clearly confirms the differential expression observed at mRNA level between the two populations. It also confirms that even within the EVP population Sox9 is only expressed at detectable protein level in less than 40% of cells (page 5-6; line 134-148) (revised Figure 1c and supplemental figure 1c, revised results)

If SOX9 distribution would be more widespread in the endothelium, the conclusion from the lineage tracing experiments in wound healing would be in doubt. The true endothelial distribution could be more strictly tested in a Sox9-CRE lineage tracing model.

As suggested by the reviewer, we have now also conducted Sox9-Cre/ER driven reporter experiments (Figure S2B). Although, we can clearly demonstrate nuclear cre expression upon tamoxifen injection, we could not observe adequate YFP expression, clearly indicating that this mouse model could not be used for lineage tracing (page 6 line-150-152). However, as a reporter, this model further confirms the expression of Sox9 in the endothelium. As discussed above, this expression seemed restricted to some of the endothelial cells and not all endothelial cells (page 6; line 149-151). However, the limitations of this model did not allow us to pursue this analysis further. This is now added in supplemental figure 2b as well as in the corresponding result section (page 6;

line 145-152).

I accept the authors' view that loss of SOX9 elevates RBPJ expression, and conversely, loss of RBPJ leads to enhanced SOX9 expression, which is in its own right an interesting finding. The postulated reciprocal negative regulation between SOX9 and RBPJ would however implicate opposite phenotypes in the EPV/D cell transition and wound healing settings. Notably, however, in both the SOX9 and RBPJ KO models, there is a depletion in EPVs, and the authors do not provide a satisfactory explanation for this, which is a bit disappointing, as this is critical for their "story". Furthermore, the response of endothelial quiescence genes (IL33, p16, p21, p57) was similar in both KO models.

We thank reviewer 1 for this comment that has also intrigued us until now. We have now explored further the negative feedback interaction of Sox9 and RBPJ and provide additional evidence for it in vivo and in vitro (see reply to reviewer 2 point 2). Most importantly we have now modified our flow cytometry protocol to allow visualisation of all possible phenotypes emanating from endothelial cells: EVP, D but also mesenchymal (page 8; line 199-209) (Supplemental Figure 3). These new settings allow to show that in the aorta, all three KO models result in different cell fate despite all resulting in reduced EVPs (page 11-12; line 312-322) (Figure 5b). Especially RBPJ eKO in the endothelium results in the most dramatic upregulation of D and mostly mesenchymal cells. Similarly, Ptch1 eKO mice result in an increase in mesenchymal cells whereas, Sox9 conditional ablation does not affect this population in the homeostatic aorta (page 8, 12-13; line 210-214, 341-349) (as opposed to skin wounds, supplemental Figure 4b).

This clearly indicates that the loss of progenitor function is affecting self-renewal and quiescence and its related genes in a similar way but is followed by different cell fates according to the gene deleted. This is now more clearly explained in the results and discussion sections. (page 15; line 418-426)

As a follow up on the comment above, it is a bit surprising to see that there are no data on the wound healing effects of RBPJ, in particular as the authors have already generated the necessary mouse crosses to conduct the analysis. It would have been interesting to learn about the effect of RBPJ ablation in the vasculature using their skin wound model in the light of their hypothesis above.

Similarly (and which would not even require the transgenic mice), it would have been nice to see siRNA experiments for RBPJ in parallel with the SOX9 siRNA experiments, which appear to give a rather distinct phenotype. Without these experiments, I am worried that the SOX9-RBPJ hypothesis is not fully validated.

We appreciate reviewer's comment. All work related to wound scarring with RBPJ eKO has already been published by us and was the initial trigger for the present study (page 9; line 252-254). As much as this work would have been useful to reproduce here to contrast the findings of Sox9eKO mice, this cannot be done. In this previous work we have detailed the endothelial to mesenchymal phenotype of RBPJeKO compared to controls as well as its functional consequences in terms of fibrosis. In the current work, we detail the relationship to Sox9 as well as the consequences of RBPJeKO on endothelial fate decisions in terms of EVP, D or mesenchymal fate (page 10; line 265-281).

There are also a number of shortcomings on the Notch side:

The authors equate Hes and Hey regulation with regulation of the Notch response. Additional markers (Nrarp, cMyc) etc could have been used.

The authors report upregulation of RBPJ as well as Hes/Hey in the SOX9-deficient cells. It is not a given that Hes and Hey will increase because of more RBPJ in the cell. In the Notch field, there are ample examples that elevated RBPJ can lead to repressed Notch activation, as RBPJ is a repressor when Notch is not activated. This would need to be discussed in more detail.

We agree with reviewer that RBPJ is not a simple effector of Notch signalling. We have attempted at examining directly some of the members of the Notch pathway to evaluate canonical Notch activation (page 10; line 265-281) (Figure 5b and c). NICD was upregulated in Sox9eKO mice suggesting that beyond an increase in RBPJ expression there is proper Notch signalling occurring in this context (page 15; line 427-429). This new evidence is not further discussed.

Minor comments:

In a number of places, there are comments referring to previous literature but with no matching citation, e.g. “SOX9 in this pathway classically attributed to TFBF signaling” (line 227). Improved referencing would be warranted.

We apologise for this omission. This reference has now been added (page 9; line 249).

The Discussion is generally a bit “all over the place”, discussing in general terms a broad set of subjects rather than focusing on discussing at depth the data in the report. As an example, the SOX9-Notch link could have been discussed in the context of previous reports where SOX9 has been shown to be a downstream genes in the Notch pathway (see e.g. data from Sean Morrison’s lab (this reviewer is also not Morrison). A better review and discussion of the existing literature would make the Discussion more interesting.

The effect of Notch signalling is very context dependent and it is difficult to establish clear links with previous literature. However in line with reviewer’s comment we have changed and reorganised our discussion with a better focus on the Sox9-Notch link (page 15; line 418-432).

The authors frequently use terms like “strikingly”, “importantly”, “we clearly observed”. These exclamatory remarks gives a “tabloid” ring to the text, which is unnecessary; the data are clear anyway.

We apologise for this language and have modified the text accordingly.

--

Reviewer #2 (Remarks to the Author):

In this manuscript, the authors utilize knockout mouse models to elucidate the role of the transcription factor Sox9 in regulating EndMT (Endothelial to Mesenchymal Transition) during skin wound healing. Based on RNA-Seq data published in a previous study (Prudence D et al., 2019), the authors find and confirm that Sox9 is highly expressed in EVP (Endovascular progenitors) in homeostasis and wound healing. Knocking out Sox9 specifically in the endothelium, using the Cdh5-CreER driver, led to a reduction in the number of EVP and an increase in D cell number. This was accompanied with both impaired EndMT and vascularization, resulting in accelerated wound healing and less scarring. To delineate the mechanism of Sox9 regulation of EndMT and its connection with different signaling pathways, the authors use additional knock-out mouse models to first, target Rbpj and Notch signaling and second, activate Hh signaling in the Cdh5+ endothelial cell population. In this manuscript, the authors provide several interesting insights for the mechanisms that regulate EndMT and their role in wound healing and scar formation.

Major comments:

1. The authors argue that Sox9 regulates EVP maintenance and represses their differentiation to D cells. Deletion of Sox9 in the endothelium reduces the likelihood of both endothelial vascularization and EndMT process, resulting in decreased scarring. From qRT-PCR results of the sorted EVP, the authors find that several genes related to Notch signaling are significantly upregulated. The RbpjeKO shows a similar phenotype of decreased EVP cell number as well as increased D cell number. The deletion of Rbpj in endothelial cells promotes EndMT and scarring but does not induce vascularization. If Sox9 and Rbpj interact via a negative feedback loop, how do the authors explain that deletion of Rbpj gene does not cause more vascularization?

We thank reviewer for this observation. We now more clearly show that despite having a reduction in EVP numbers and self-renewal, each conditional KO has a distinct phenotype in terms of endothelial cell fate (page 11-12; line 316-326). We have now modified our flow cytometry protocol to allow visualisation of all possible phenotypes emanating from endothelial cells: EVP, D but also mesenchymal (page 8; line 199-209) (Supplemental Figure 4). These new settings allow to show that in the aorta, all three KO models result in different cell fate despite all resulting in reduced EVPs (page 11-12; line 312-322) (Figure 5b). Especially RBPJ eKO in the endothelium results in the most dramatic upregulation of D and mostly mesenchymal cells. Similarly Ptch1 eKO mice result in an increase in mesenchymal cells whereas, Sox9 conditional ablation does not affect this population in the homeostatic aorta (as opposed to skin wounds, supplemental Figure 4b) (page 8, 12-13; line 221-226, 343-349).

This clearly indicates that the loss of progenitor function is affecting self-renewal and quiescence and its related genes in a similar way but is followed by different cell fates according to the gene deleted. This is now more clearly explained in the results and discussion sections. (page 15; line 418-426)

2. When the authors analyze gene expression differences in the Sox9 KO mice, they use sorted EVP cells in both control and Sox9 KO, but when comparing the differences in RbpjeKO mice they use the whole endothelial population. The observed upregulation of Notch signaling in EVP cells from Sox9eKO mice is thus not directly comparable to the

upregulation of Sox9 in whole endothelial cells. Therefore, this weakens the argument that Sox9 and Rbpj interact via a negative feedback loop in regulating EndMT process. Additionally, both knockout mice show the similar trend in decreased number of EVP cells. **We agree with reviewer that it has been difficult to perform studies on a homogenous population across the three conditional knockout models versus controls. In particular sorting equivalent populations in sufficient quantities is not possible from all mouse models as highlighted in Figure 6b given the specific effects of each knockout on cell populations. We have therefore performed a new series of experiment to better characterise the negative feedback interaction of Sox9 and RBPJ and provide additional evidence for it in vivo and in vitro (page 10, 15; line 265-281, 418-426).**

We now provide evidence that in vivo, the aortic endothelium of Sox9eKO animals expresses higher levels of nuclear NICD suggesting an activated canonical notch signalling at the protein level. In contrast, aortic endothelium from the RBPJeKO animals express higher levels of nuclear Sox9 at protein level. (page 10; Line 265 – 272)

We have also conducted experiments in vitro to show that gamma-secretase inhibition reducing canonical Notch signalling in primary culture ECFCs results in an increase in Sox9 gene expression. (page 10; line 272-281)

We believe this additional evidence more strongly corroborates our claims and have now discussed this further in the discussion section.(page 15; Line 418-426)

3. The authors propose that Hh signaling plays a critical role in the EndMT process. The authors start to compare gene expression in CD34+ and CD34- populations in both control and Ptc1eKO mice. They show that Sox9 is expressed in EVP (CD34+) cells but surprisingly they also show that Sox9 is expressed in CD34- cells as well. This is important and was not mentioned previously in the manuscript. The authors did not perform a similar analysis of Sox9 expression in the Sox9eKO and RbpjeKO mouse models, to test whether there are differences in gene expression between CD34- and CD34+ cells.

It is important to note that in all these experiments only the YFP+ cells are considered. This ensures that only cells that were endothelial in origin and expressing cdh5 (VE-Cadherin) have been examined. As explained above, sorting YFP+ and CD34- populations in Sox9eKO is difficult due to low numbers and in RBPJeKO, the majority of cells are CD34- (page 11-12; line 312-322).

Furthermore, their results show that in the Patch1eKO mice the expression pattern of the endothelium (CD34+) resembles that of the Sox9eKO but also that CD34- cells have similar expression (Sox9 up, Notch down) to the endothelium in RbpjeKO. The scarring phenotype is similar between Patch1eKO and RbpjeKO. Does this mean that the CD34- population takes over the role of CD34+ population-EVP cells and becomes the key population for EndMT to promote scarring? The authors provide complicated data but they do not identify clearly what is CD34- population and whether Sox9's function in EndMT process is intergrated with or regulated by Notch signaling or Hh signaling.

We acknowledge the point raised by reviewer around the identity of CD34- populations in the previous version. In our revised manuscript we have now performed a new flow cytometry panel allowing to gate on all Lin-negative (non-hematopoietic) and YFP+ endothelial cells and then examine the expression of this YFP+ population across CD31 and CD34 (Figure S3A) (page 8; line 199-209).

We can therefore show that the CD34- fraction is actually CD31-CD34- and increased in quantity in ptch1eKO and rbpjeKO mice (page 12; line 318-322). The gene expression in this population as opposed to the EVP and D populations is reminiscent of EndMT. We provide evidence that this population expresses CD26, PDGFRa and aSMA at higher levels compared to fully differentiated endothelial cells (D population) (page 8; line 205-209). This new characterisation further reinforces the role of Sox9 in the EndMT process occurring in the Ptch1eKO mice. Sox9 is only upregulated in the population undergoing this transition. We therefore agree with reviewer that this population is the most likely contributor to EndMT and have therefore named it mesenchymal. This is now also a new part of the discussion section (page 15; Line 418-426).

4. Previous studies have shown that Sox9 gene is critical for hair follicle stem cell maintenance. Conditional deletion of Sox9 in hair follicles triggers the differentiation of hair follicle keratinocytes to epidermal cells (Meelis Kadaja et al., 2014), which may facilitate the wound healing. Here the authors perform Sox9 siRNA topical treatment after wounding which may also affect Sox9 expression in hair follicle stem cells. Given the broader implications of this treatment the authors should analyze the histology of hair follicles and the kinetics of the hair cycle. Also, keratinocyte proliferation and apoptosis should be assessed after topical treatment, to make sure the phenotype is not caused by the changes of keratinocyte activity rather than the endothelium. The author mention this in the discussion but did not provide any data.

This is an interesting suggestion, although out of the scope of the present work. Of note, in line with the reviewer's comment there is an acceleration of the speed of wound healing in both the Sox9eKO mice and the Sox9 siRNA treated mice compared to controls that cannot be explained by the EndMT phenotype (page 17; line464-468). Many possibilities could be explored to explain this observation. However, this acceleration in healing indicates that the siRNA therapy against Sox9 does not impede epidermal progenitors' ability to migrate/proliferate to heal wounds.

Other issues:

1. The authors should test more mesenchymal markers, such as a-SMA, FSP-1, vimentin and Colla2, not only by qRT-PCR, but also by IF. The authors should also provide more Immunostaining data of EndMT markers, snail and a-SMA, co-staining with CD31.

We have provided markers such as a-SMA, Slug, CD31 and YFP to evaluate EndMT. We provide additional data on CD26, PDGFRa (page 8; line 206-209) as well as FSP1 in Figure 4. Of importance all markers are used as immunofluorescence.

2. The resolution of Sirius red staining data is low; it is hard to evaluate the difference between wildtype and Sox9eKO mice.

We have now provided higher magnification and resolution of the Sirius red staining in Fig 3e (page 7-8; line 196-198).

3. The authors create 1.5 cm X 1.5 cm wounds to assess fibrosis, but all the flow data are

made from 6 mm punch wounds. It is an open question whether the pattern of EVP and D cell changes are consistent in these different wound scenarios.

We have conducted all experiments in both models for immunofluorescence. Large wounds have the advantage of producing clinically visible scars in murine models that can be measured macroscopically (page 7; line 186-188). However, small 6mm punch wounds have the important benefit of having a very precise timing of events with regards to peak of angiogenesis and wound closure. We now produce additional data in small wounds to demonstrate that the scarring phenotype is also improved in small wounds (Figure 3e)(page 8, 9; line 221-226, 234-238).

4. In figure 3C, is wound healing also affected from day 4 to day 12?

As shown in figure, the healing is definitely affected but the difference did not reach significance at these timepoints.

6. The authors broadly define the CD34- population as mesenchymal cells. I feel that additional surface markers, like α -SMA, S100a4, are needed to better characterize this population.

We agree with reviewer. We have provided a new definition of these CD34- cells by showing that as M cells they are YFP+ (of endothelial origin) but have lost both CD34 and CD31, express CD26 and PDGFR α as well as α SMA (figure S4b) (page 8; line 205-209).

7. For figures 4C, 6C, 6D and 7D, the authors should also provide representative high resolution immunostaining data.

We now provide new high res images in Figures 4 and 7

8. To confirm Sox9 upregulation in the different cell populations, the authors should demonstrate Sox9 expression by immunostaining in the Rbpj Δ KO and Ptch1 Δ KO mice after wounding, not just by RT-PCR.

We thank reviewer 2 for this suggestion. This is now provided in figure 5

9. The phenotype of Rbpj Δ KO mice was published previously by the same author, so this part is less novel.

We have carefully avoided to repeat the phenotype and EndMT data reported in this previous paper. The novelty is the change in EVP, D and M proportion in this scenario as well as the aorta EVP self-renewal that has never been explored previously.

--

Reviewer #3 (Remarks to the Author):

Patel and colleagues set out to examine the role of Sox9 and Rbpj signaling on endothelial progenitor cells during tissue repair. The rationale for these studies is that endothelial-to-mesenchymal transition is a major contributor to fibrosis during skin wound healing. Although this may be true of cardiac wound healing, I am not convinced that this has been

definitively shown for skin and this manuscript in its current form does not directly support that claim any further.

As highlighted by reviewer 2, we have previously published that EndMT plays a crucial role in the reduction of endothelial cells in skin wounds after the peak of angiogenesis and contributes to skin fibrosis. We have also reported that EndMT in wounds is accelerated by RBPJ conditional deletion (Patel et al, J Invest Dermatol 2018 May;138(5):1166-1175). This previous work initiated the current study (page 4; line 92-95).

The authors employ mouse models in which they genetically delete either Sox9, Ptch or Rbpj within “endothelial progenitors” and subsequently evaluate the impact of each on quality of skin wound healing. The authors provide evidence suggesting that intriguingly, loss of Sox9 in endothelial progenitors results in reduced scar formation and this is also corroborated by topical application of siRNA against Sox9. Although interesting, the authors fail to provide definitive evidence from their fate mapping experiments. Although they provide cytometry data, there is a lack of imaging showing specificity of labeling within cutaneous endothelial progenitor cells or their fates following skin injury.

We thank reviewer for highlighting the importance of fate mapping. We have provided additional images at high resolution regarding the fate of YFP+ endothelial cells in wounds upon Sox9eKO (figure S5C). These images undoubtedly show the coexpression of mesenchymal markers such as SLUG, aSMA but also FSP1. These mesenchymal markers can be identified in cells that maintain CD31 expression and a vascular structure. But they can also be identified in cells that have lost CD31 and the vessel structure, being identified as individual cells in the granulation tissue (page 9; line 236-236).

In addition we have improved our definition of mesenchymal cells in flow cytometry through the identification of YFP+ cells that are devoid of CD34 or CD31 but express CD26, PDGFRa or aSMA.(figure S4b) (page 8; line 206-209)

It would also be important to show that these same genes (Sox9, Ptch, Rbpj) are co-expressed in the putative endothelial progenitors in vivo during homeostasis and following injury. Although I think the manuscript is quite interesting and would be of importance to the wound healing community, there are a number of shortcomings that need to be addressed (described below).

We thank reviewer 3 for this suggestion. We now provide evidence in aorta that the same cells can coexpress RBPJ and Sox9 especially in sorted EVPs during homeostasis (figure 1c)(page 5-6; line 134-138). In wounds, given the reciprocal negative regulatory feedback as shown along the manuscript, we suggest that such occurrence of co-expression is less likely.

1. Figure 1 – The authors need to show confocal images of skin sections from Cdh5Cre mice that demonstrate a) exclusive expression in endothelial cells, but not other dermal cell types, b) co-expression with Sox9 and c) verification with additional endothelial markers.

We agree with reviewer that this is an important first step that is now highlighted in supplemental figure 2A. Immediately post-induction all YFP+ cells in wounds, aorta or

normal skin are CD31+ and endothelial. This clearly shows the specificity of the initial labelling of endothelial cells (page 6; line 144-147).

Moreover we have now shown co-expression of Sox9 and CD31+ cells in the normal skin and in the aorta.(page 5; line 118-121, Figure 1A, B)

I do not understand the rationale for doing in vitro immunostaining of cells that have been isolated from acute skin wounds.

If the authors are going to claim that “Sox9 expressing endothelial progenitors” are contributing to skin wound healing then they need to demonstrate 1) their presence in normal skin vasculature and 2) then show active recruitment into the wound (and differentiation to myofibroblast phenotype within the wound).

We apologise for this misunderstanding. We have now provided evidence of SOX9 expression in the normal skin vasculature in Fig1B. Moreover Fig 1D shows a D1 wound section where individual YFP+ EVPs can be labelled with SOX9 in vivo. This is not a sorted cell (page 23; line 652). We have therefore further clarified the figure legend.

Sox9 expressing endothelial cells are therefore present and contribute to wounds from D1 post-wounding (page 6; line 146-149).

Providing a kinetic analysis at different times post-injury would be an essential step in defining their contribution to wound healing and active contribution to scar formation/neodermis.

In previous studies we have shown the kinetics of endothelial population infiltration of skin wound granulation tissue (Patel et al Circulation 2017). In particular we have shown that Sox18 expressing EVPs infiltrate the wound as single cells as early as D1. We now provide additional evidence that these cells express Sox9 (page 6; line 146-149). Moreover in our previous work, we showed that TA and D cells only appear from D3 and EndMT can only be detected after the peak of angiogenesis (D5). We agree with reviewer that providing detailed data of Sox9eKO at different time point is important. We have therefore provided additional flow cytometry data at D5 and D7 post-wounding (page 8; line 210 – 218) (fig S5B). These findings show similar patterns of reduction in EVP and D cells over time and the increase in EndMT. Sox9eKO wounds harbour less EndMT and more YFP+cells maintain an endothelial phenotype (page 9; line 229-238).

2. “Additionally, skin wound analysis from Sox9 conditional knock-out mice demonstrated a significant reduction in pathological EndMT resulting in reduced scar area.”

a. What was the impact on myofibroblast numbers in the wound? This needs to be quantified.

Although the number of YFP+ aSMA+ was divided by 2 upon Sox9 ablation in the endothelium (Figure 4), we did not observe a significant change in global aSMA expressing cells. This might be due to the fact that many myofibroblasts do not have an endothelial origin and therefore reduce the difference between Sox9eKO and controls. The relative contribution of myofibroblasts of endothelial and non-endothelial origin to

fibrosis is unknown but based on our findings, one could speculate that myofibroblasts of endothelial origin play a crucial role in providing about 30% of the total scarring.

b. How do you know that you haven't just accelerated/increased contraction of the wound rather than actually mitigating fibrosis? Can you measure overall contraction of the wound in each genotype? This is particularly important given that the mouse skin healing is primarily driven by contraction and you have not splinted the wounds to specifically look at secondary intention. Histological images of the wounds would be helpful to understand the overall impact on wound outcomes. Inclusion of second harmonic imaging to characterize the collagen content/orientation would also be useful. Cd26 staining alone is not indicative of a 'fibrotic' fibroblast.

We agree with reviewer that murine wound healing is strongly dependent on contraction. Wound contraction also importantly depends on myofibroblasts and as discussed above we did not observe a significant change in total myofibroblast numbers. Moreover Sox9eKO mice produced less myofibroblasts of endothelial origin and are therefore less likely to contract wounds (figure 4C).

We now provide additional images of Sirius red staining and quantification to further show the reduction in wound scarring (figure 3E).

3. It is not clear to me why you are examining "human placental fetal ECFC and MSCs" – this seems entirely extraneous. Given the focus is on skin wound healing, the authors should focus their analysis to the vascular endothelial cells within the skin.

In vascular biology, ECFCs from the cord blood are the gold standard in vitro model of endothelial progenitors as recently defined in a consensus statement (Medina et al, Stem Cells Transl Med. 2017 May;6(5):1316-1320.). Placental ECFCs have been shown to be equivalent to cord-blood ECFCs and emanate from the same fetal origin.

The use of well recognised progenitor populations with self-renewal is important in the context of our work as opposed to HUVECS or dermal endothelial cells where self-renewal and plasticity is poorly defined.

4. Similarly the analysis of endothelial cells from aorta in this manuscript is again of little relevance. All of the analyses currently performed using aorta should be done on cutaneous endothelial cells.

We agree with reviewer that in principle the skin endothelium would be interesting to study. The small number of cells obtained from the skin is a technical challenge that impedes the use of this tissue. However, aortic endothelial cells can be obtained without major technical hurdle and the hierarchy described by us previously is present in both skin and aorta. To establish progenitor function, in comparison to previously reported assays, the use of aorta is justified.

5. When do the EPV (or their derivatives) appear within the skin wound? Where do they go? How many of them relative to mobilized dermal fibroblasts? Showing kinetics and images of cells within at different times following injury is essential to underscore the importance/contribution of these cells to the wound healing process.

We apologise for not clarifying this in the introduction (page 3; line 79-83). We have previously reported that EVPs are the first cells infiltrating the skin wound granulation

tissue at D1. We have reported their kinetics in absolute numbers and have performed lineage tracing with Sox18-Cre and cdh5-Cre models to demonstrate that they form TA and D cells. In wounds, D cells appear only from D3 and increase in numbers until D5. From D5 the total number of endothelial cells goes down and no more EVPs are found at D7. From D7 to D14 there is a steady decline in endothelial cells in part explained by EndMT. These details have been already published by us in Patel et al, Circulation 2017 and Patel et al, J Invest Dermatol 2018. We have now expanded on these previous findings in the introduction section of the revised manuscript.

6. As it stands, the title of the paper is not well supported by the data presented.

We respectfully disagree. We have provided ample evidence of the change in phenotype upon loss of Sox9 of endothelial cells avoiding a mesenchymal transition using 5 different markers (page 8, 9; line 203-209, 227-240) (αSMA, FSP1, SLUG, CD26, PDGFRα). We have also provided evidence that in other models where Sox9 is upregulated, more EndMT occurs (page 9, 11-12; line 265-281, 286-288, 312-322). Finally we have provided evidence that siRNA treatment of wounds reduces EndMT and scarring. We hope the added evidence highlights the importance of Sox9 in driving EndMT in this revised version.

7. Why have you chosen to use two different skin wound models (6mm diameter excision versus 1.5 x 1.5cm)? Using a wound size of 1.5 x 1.5 cm, it would be expected that there would be some level of hair follicle neogenesis. Was this observed? This might be a better proxy for reduced fibrosis and increased regeneration in any of your genetic KO models.

The 1.5x1.5 wounds were not used for hair follicle neogenesis but for the measurable macroscopic scars they leave behind. This ensures that the effect of Sox9 deletion in the vasculature has clinical relevance and results in measurable reduction in scar size (page 7; line 186-188).

8. Is there a difference between the EPV contribution to your 6mm wound and 1.5cm wound ?

We have not observed a difference in hierarchy and EVP, TA and D cells are present in large wounds. The small 6mm wounds present the advantage of a well timed and reproducible set of outcomes. We have now also shown the differences in scarring in small wounds ensuring there are no differences in outcomes between large and small wounds.

9. The measures used for fibrosis are quite limited (e.g. picrosirius red quantification and Cd26 – presence of cd26 in skin is not definitive of a fibrotic phenotype given it is expressed in uninjured papillary dermis and hair follicle mesenchyme). This should be expanded upon. How many sections per wound are you measuring across the wound and from where are you sampling? This needs to be made clear. Inclusion of second harmonic generation to measure collagen content and orientation, dermal thickness, dermal cell density would be much more convincing.

We agree with reviewer and have used CD26 as another mesenchymal marker and do not consider it as a measure of fibrosis. Our main measure of fibrosis are the size of the scar in large wounds as well as the picrosirius red quantification that has been

performed on 3 sections per wound (sections through the centre of the wound/scar) (page 31; line 846-847). We strongly believe that the macroscopic scar size reduction is by far the best indicator of the clinical outcome to achieve and the Picrosirius red provides a quantification of the collagen content. This is now clarified in results section in Fig3e.

Text comments:

Lines 123-128 : Please show the FACS strategy in the figures.

The FACS strategy is detailed in supplemental figures 3 and 4 for both strategies employed in the revised manuscript.

Lines 129-130 : “We have reported the accumulation of EVP” – Can you please describe the relevance of this statement?

This sentence has been changed to: We have reported the presence of EVP... This recapitulates our findings of EVP in the granulation tissues of wounds at D1 (page 6; 139-141).

Line 132 : Cdh5 : show in skin to trust the specificity of Cadherin 5 in skin or ref ?

This is now clarified in supplemental figure 2a

Line 135 : What is the center of the wound at D1 ? How is it possible to have cells in the center of the wound, except immune cells and blood ? Images of the skin wounds at Day 1 need to be shown to verify labeled cells in the centre of the wound.

To avoid duplicating the data already published, we would refer to our previous publication where D1 wounds are largely explored with immunostaining showing the distribution of EVPs as well as their surface markers

Lines 138-141 : add a schematic to explain the different populations

Lines 157-158 : It would be helpful to add a schematic to explain EPV, D, TA and FACS strategy

We have now added a new gating strategy to clarify this point in Supplemental Figure 4

Lines 166-175 : How did you assess that ? Brdu ? EdU ? When did you inject the DNA labeling ? This does not appear to be described in the material & methods.

We apologise if this does not appear in methods. The cell cycle was established based on DNA labelling. We have now corrected this in methods (page 33; line 847-851) and figure legends.

Line 186 : Picrosirius red staining: the staining doesn't seem appropriate, as if the tissue sections are folded. Shouldn't you see 3 distinct skin layers within the normal skin? Higher magnification images should be included to verify the rigor of the quantification.

We have now provided higher magnification images of the picrosirius red on serial sections of wounds.

Lines 199-201 : Explanation for inclusion of SLUG staining should be made clear. Why are you not showing aSMA images along with the SLUG ?

Lines 204-207: show higher magnification images. What is the significance of CD34+/-, CD31+/-?

We have provided further references and reasons behind for the inclusion of SLUG (page 16; line 454-457). Activation of SLUG does not directly induce EndMT to a mesenchymal cell type that is α SMA positive. Higher magnification images are now provided in figure S5C. We have now modified our flow cytometry protocol to allow visualisation of all possible phenotypes emanating from endothelial cells: EVP, D but also mesenchymal cells CD31-CD34- (Supplemental Figure 3)

Lines 357-358 / 362-364 : “blood vessel formation or to enter Mesenchymal transition”. Unfortunately, you never show that. If you want to keep this title and this interesting conclusion, you should show a skin wound with YFP+ cells into the dermis, not in the vessels and collagen productive cells, for example

We thank reviewer for this suggestion, We have provided examples of this observation of single YFP+ cells stained with mesenchymal markers (figure S5C).

Lines 784 – 785: should be μ and not uM

We apologies for this inconsistency, and this mistake is now corrected

Lines 867-879: you should explain which method you use for qPCR quantification **method for the quantification of qPCR gene expression changes is now included in materials and methods (page 34; line 866-867)**

General figure comments:

1. Your Y labels are not clear at all in all the figures (percentage of what ? fold change of what % of gated ?)

Figures legends are changed accordingly to reflect fold- and percentage- changes.

2. For all of your qPCR controls and the percentage of scar area you don't include an indication of variance (SEM or SD)? Please show this!

We apologies for this and have corrected and added the indication of variance within the figure legends.

3. Please add n-values for # of mice used in each experiment and indicate this for each analysis in the figure legends.

This is corrected and added to each respective figure legends

Specific comments on figures :

- Figure 1: Not clear if we are in skin / aorta ? Please add a schematic

Organ bed for image displayed is now added to the figure

- Change the order in the figures 4D and 4E-F

Figure 4 is now focused on EndMT in wound healing

- Figure 3-C : the wound closure in LOW is done around at D14-D16. How do you evaluate this percentage?

The percentage is calculated based on the initial wound size.

- Fig 3D,E – The imaging provided is entirely unconvincing. How can you actually measure anything from these images? Additional images and details of how this was exactly quantified will need to be provided.

Additional PSR staining is provided for collagen quantification

- Figure 3E: high magnification or better image?

Better images are now provided in Fig 3E

- Figure 4-C: Why do these different vessel signatures are different?

Vessel signatures are now quantified via FACS in Fig S5B whereby the total endothelial population within the D5 wound is not affected by Sox9 knockout

- Fig S6 : should be in the main figures

To avoid data duplication, as these results were previously published (Patel et al, J Invest Dermatol 2018 May;138(5):1166-1175) Therefore it was omitted in the result figures.

- Fig S1 : Figure S1B “We also observed that the SOX9 staining is localised only to the endothelial layer and not in the underlying vascular smooth muscle layers “ – What is the red staining? Why are you not showing images of Sox9 labeling in the deeper image slices? This set of panels is confusing and needs to be redone.

We apologise for the confusion. Markers used within this image is now provided. The endothelium was stained with CD31 (red) and SOX9 (yellow), and the 3D z-stack demonstrated exclusive SOX9 expression in the endothelium. Underlying slices of vascular smooth muscle cells did not express SOX9 (page 5; line 119-122).

- S2A: This is surprising that you don't see anything in the KO – can you explain this?

We could not successfully amplify *Sox9* with qPCR from endothelial cells isolated from SOX9^{eKO} mice

- S2B. Why are you not showing YFP fluorescence?

YFP expression in *en face* whole mount preparation does not provide clear outlines for individual endothelial cells compared to junctional proteins such as CD31.

Reviewers' Comments:

Reviewer #1:

Remarks to the Author:

The authors have satisfactorily addressed the critique on the original version. The ms is recommended for acceptance.

Reviewer #2:

Remarks to the Author:

The revised manuscript by Zhao et al. is significantly improved. The authors should be commended for their effort in performing new complementary experiments and providing new data to address some of the issues that were raised in the first round of review. In particular, the new flow cytometry strategy that better characterized the mesenchymal population, painting a more complete picture of the diverse phenotypic effects resulting from each genetic manipulation. For the most part my comments were sufficiently addressed.

Major comments:

1. The authors response to my comment is satisfactory.
2. The authors response to my comment is satisfactory.
3. The authors response to my comment is satisfactory.
4. I partially agree with the authors response to my comment. According to Ge & Fuchs. (Cell, 2017) in their working model for lineage infidelity and demonstrate an important role for epithelial-expressed Sox9 in wound healing. This may not be relevant to the Sox9eKO experiments but may contribute to the phenotype of the Sox9 siRNA experiment. I understand that these experiments are not essential for the main conclusions of this study and that due to Covid19 and other factors my comment could not be accommodated, and I can fully understand the authors predicament.

Other issues:

1. The authors response to my comment is satisfactory.
2. The authors response to my comment is satisfactory.
3. The authors response to my comment is satisfactory.
4. I partially agree with the authors response to my comment. Their statement is contradictory, in that lack of statistical significance indicate no difference in the early stage of wound healing until day 12.
5. (please excuse my typo and the lack of bullet 5 in my initial review)
6. The authors response to my comment is satisfactory.
7. The authors response to my comment is satisfactory.
8. The authors response to my comment is satisfactory.
9. The authors response to my comment is satisfactory.

In addition to my previous comments, I strongly believe that this manuscript would benefit tremendously by the inclusion of a graphical diagram which would clearly illustrate their working model based on the experimental data from this study. This would assist the non-expert reader, especially given the broad appeal of this journal. Furthermore, I would suggest that the authors consider re-working the layout of their figures as some panels are too small to be easily legible.

In conclusion I believe this study is sufficiently novel and provides significant insight into the molecular mechanisms of endothelial to mesenchymal transition in wound healing.

Reviewer #3:

Remarks to the Author:

I would like to thank the authors for their clarifications and additional data they have provided. Although I would prefer a more rigorous analysis of scar quality (second harmonic imaging, persistence of aSMA expressing cells, dermal thickness) rather than just size, I believe they have done a reasonable job in addressing the previous reviewer suggestions.

There are only two minor queries that may require followup:

1. Figure 1b colocalization of Sox9 and endothelial cells. I still find the data presented in this panel troubling- Why do all other examples (aorta) and hair follicle epithelial cells show defined nuclear labeling for Sox9, while these skin endothelial cells seem to have a relative exclusion of Sox9 from the nucleus? These images are still unconvincing in demonstrating co-localization of the endothelial marker (CD31, red) with Sox9. Hence it seems possible that it is expressed in another vascular associated cell type that is not endothelial. Higher magnification confocal images with inclusion of z planes would help to address this.

2. In figure 3e, I would remove the initial picrosirius staining provided in the upper panels and only include the lower panels (new images) as the latter is much more convincing of the quantification data and the conclusions you have drawn.

3. In figure S5, arrows should be included in all panels to indicate the location of the representative cells that are exhibiting co-localization. As well some of the panels do not seem to align appropriately with the merged image. (cropping issue?)

REVIEWERS' COMMENTS

Reviewer #1 (Remarks to the Author):

The authors have satisfactorily addressed the critique on the original version. The ms is recommended for acceptance.

We thank reviewer 1 for all the excellent suggestions that have greatly improved our manuscript.

Reviewer #2 (Remarks to the Author):

The revised manuscript by Zhao et al. is significantly improved. The authors should be commended for their effort in performing new complementary experiments and providing new data to address some of the issues that were raised in the first round of review. In particular, the new flow cytometry strategy that better characterized the mesenchymal population, painting a more complete picture of the diverse phenotypic effects resulting from each genetic manipulation. For the most part my comments were sufficiently addressed.

We would like to express our sincere thanks to Reviewer 2 and the comments that made our work more focused and convincing.

Major comments:

1. The authors response to my comment is satisfactory.
2. The authors response to my comment is satisfactory.
3. The authors response to my comment is satisfactory.
4. I partially agree with the authors response to my comment. According to Ge & Fuchs. (Cell, 2017) in their working model for lineage infidelity and demonstrate an important role for epithelial-expressed Sox9 in wound healing. This may not be relevant to the Sox9eKO experiments but may contribute to the phenotype of the Sox9 siRNA experiment. I understand that these experiments are not essential for the main conclusions of this study and that due to Covid19 and other factors my comment could not be accommodated, and I can fully understand the authors predicament.

As pointed by the reviewer, this is an exciting question that is partially answered through the use of a conditional KO. Beyond this paper further research about the impact of Sox9siRNA on other cell types in the wound will address this issue.

Other issues:

1. The authors response to my comment is satisfactory.
2. The authors response to my comment is satisfactory.
3. The authors response to my comment is satisfactory.
4. I partially agree with the authors response to my comment. Their statement is contradictory, in that lack of statistical significance indicate no difference in the early stage of wound healing until day 12.

This is correct and has been acknowledged in the manuscript, line 180.

5. (please excuse my typo and the lack of bullet 5 in my initial review)
6. The authors response to my comment is satisfactory.
7. The authors response to my comment is satisfactory.
8. The authors response to my comment is satisfactory.
9. The authors response to my comment is satisfactory.

In addition to my previous comments, I strongly believe that this manuscript would benefit tremendously by the inclusion of a graphical diagram which would clearly illustrate their working model based on the experimental data from this study. This would assist the non-expert reader, especially given the broad appeal of this journal. Furthermore, I would suggest that the authors consider re-working the layout of their figures as some panels are too small to be easily legible.

In conclusion I believe this study is sufficiently novel and provides significant insight into the molecular mechanisms of endothelial to mesenchymal transition in wound healing.

We thank the reviewer for this suggestion, and have provided a schematic summary illustrating our findings. Also we have re-organised some of the figures to increase the size of photomicrograph panels.

Reviewer #3 (Remarks to the Author):

I would like to thank the authors for their clarifications and additional data they have provided. Although I would prefer a more rigorous analysis of scar quality (second harmonic imaging, persistence of aSMA expressing cells, dermal thickness) rather than just size, I believe they have done a reasonable job in addressing the previous reviewer suggestions.

We would like to thank reviewer 3 for the excellent suggestions regarding the further assessment of wounds.

There are only two minor queries that may require followup:

1. Figure 1b colocalization of Sox9 and endothelial cells. I still find the data presented in this panel troubling- Why do all other examples (aorta) and hair follicle epithelial cells show defined nuclear labeling for Sox9, while these skin endothelial cells seem to have a relative exclusion of Sox9 from the nucleus? These images are still unconvincing in demonstrating co-localization of the endothelial marker (CD31, red) with Sox9. Hence it seems possible that it is expressed in another vascular associated cell type that is not endothelial. Higher magnification confocal images with inclusion of z planes would help to address this.

We agree with reviewer that the use of 3D whole mount imaging with stacked images cannot give convincing evidence of colocalization. We have therefore provided z planes to more clearly highlight colocalization. Moreover we have also performed additional staining on the skin to reveal nuclear staining of Sox9. These images have been added to Figure 1.

2. In figure 3e, I would remove the initial picosirius staining provided in the upper panels and only

include the lower panels (new images) as the latter is much more convincing of the quantification data and the conclusions you have drawn.

We thank the reviewer for this suggestion, this is now removed from the figure

3. In figure S5, arrows should be included in all panels to indicate the location of the representative cells that are exhibiting co-localization. As well some of the panels do not seem to align appropriately with the merged image. (cropping issue?)

We thank reviewer 3 for this comment, arrows have been added to each panel for better representation of co-localisation.